# Exploring the Limitations of Layer Synchronization in Spiking Neural Networks

## Abstract

Neural-network processing in machine learning applications relies on layer synchronization. This is practiced even in artificial Spiking Neural Networks (SNNs), which are touted as consistent with neurobiology, in spite of processing in the brain being in fact asynchronous. A truly asynchronous system however would allow all neurons to evaluate concurrently their threshold and emit spikes upon receiving any presynaptic current. Omitting layer synchronization is potentially beneficial, for latency and energy efficiency, but asynchronous execution of models previously trained with layer synchronization may entail a mismatch in network dynamics and performance. We present and quantify this problem, and show that models trained with layer synchronization either perform poorly in absence of the synchronization, or fail to benefit from any energy and latency reduction, when such a mechanism is in place. We then explore a potential solution direction, based on a generalization of backpropagation-based training that integrates knowledge about an asynchronous execution scheduling strategy, for learning models suitable for asynchronous processing. We experiment with 2 asynchronous neuron execution scheduling strategies in datasets that encode spatial and temporal information, and we show the potential of asynchronous processing to use less spikes (up to 50%), complete inference faster (up to 2x), and achieve competitive or even better accuracy (up to ∼10% higher). Our exploration affirms that asynchronous event-based AI processing can be indeed more efficient, but we need to rethink how we train our SNN models to benefit from it.

## 1 Introduction

Artificial Neural Networks (ANNs) are the foundation behind many of the recently successful developments in AI, such as in computer vision Szegedy et al. (2017); Voulodimos et al. (2018) and natural language processing Vaswani et al. (2017); Brown et al. (2020). To match the complexity of the ever more demanding tasks, networks have grown in size, with advanced large language models having billions of parameters Zhao et al. (2024). With this, the power consumption has exploded Luccioni et al. (2023), limiting the deployment to large data centers. In an effort to learn from our brain's superior power efficiency, and motivated by neuroscience research, SNNs Maass (1997) bolster as an alternative. They use discrete binary or graded spikes (events) for communication, are suited for processing sparse features He et al. (2020), and when combined with asynchronous event-based processing are assumed to enhance latency and energy efficiency. Sparsity leads to fewer synaptic operations, resulting in low energy consumption, and asynchronous operation potentiates concurrent evaluation of all neurons in the network purely event-driven, leading to low latency.

Conventional highly parallel ANN compute accelerators, such as Graphics Processing Units (GPUs) and Tensor Processing Units (TPUs), which are primarily optimized for dense vectorized matrix operations, face inherent challenges in exploiting unstructured and temporal sparsity for improving their energy efficiency. Targeting the common practice of executing ANNs layer-by-layer has left them with poor support for asynchronous processing (for improving latency). At best, they parallelize processing within a layer and/or pipeline processing across layers. This leaves an exploration space for neuromorphic processors that try to excel in handling the event-driven nature of SNNs and leverage asynchronous concurrent processing, offering efficiency advantages in various tasks Ivanov et al. (2022); Kang et al. (2020a); Müller-Cleve et al. (2022).

However, despite this advancement, the training of SNNs today very often conveniently relies on conventional end-to-end ANN training methods for performance Dampfhoffer et al. (2023), which organize/synchronize computations per-layer rather than event-driven per neuron Guo et al. (2023a). Specifically, at any given discrete timestep within a neuron layer, first, the total of all presynaptic currents (from the preceding layer) must be computed and integrated, before postsynaptic neurons in the current layer update their state and evaluate their activation function (i.e. emitting new spikes). For consistency, neuron evaluation in one layer must thus complete and synchronize before proceeding to evaluate neurons of a next layer. This *breadth-first* processing approach (with per-layer synchronization), while it facilitates the use of vectorized computing hardware (such as GPUs) during training, introduces dependencies on per-layer synchronization that could impact model accuracy if altered during inference. To avoid this situation, even asynchronous neuromorphic processors, such as Loihi Davies et al. (2018), have integrated mechanisms to ensure (and enforce) layer synchronization.

This leaves a crucial (efficiency) aspect of SNNs relatively unexplored: the ability to allow spiking completely asynchronously across the network without having separate phases for integration and firing, just like in our brain Zeki (2015). In that neurophysical modus operandi, neurons can fire and receive currents anywhere in the network at any time, completely independent, a concept we term *"network asynchrony"*. Allowing network asynchrony can be advantageous Pfeiffer & Pfeil (2018), as we confirm in our results. Spike activity can quickly propagate deep into the network without being bound by synchronization barriers, thus reducing latency. Furthermore, adhering to layer synchronization could lead to increased computational overhead as the network scales, as suggested by Amdahl's law Rodgers (1985). This implies that the overhead grows non-linearly by adding more computational units to a group that needs to be synchronized at some point in time Yavits et al. (2014). With network asynchrony, such groups can be kept smaller, and the number of synchronization moments can be minimized, potentially reducing the waiting time.

In this paper, we demonstrate this problem and explore solutions. Using a simulation environment that implements the concept of network asynchrony, we provide quantitative results on benchmark datasets (with different spatio-temporal information content) and network topologies of two or more hidden layers, which show the performance degradation and latency/energy inefficiency resulting from changes in model dynamics when trained with layer synchronization and later deployed for asynchronous inference. Next, we explore a potentially promising solution by proposing a generalisation of gradient (backpropagation-based) training, that can be parameterized with various neuron execution scheduling strategies for asynchronous processing, and vectorization abilities present in various neuromorphic processors. We show that using this training method, it is possible not only to recover the compromised accuracy, but also to fulfil the expectation of saving energy and improving latency under asynchronous processing (when compared to the conventional breadth-first processing in GPUs). This work opens a path for design space explorations aimed to bridge the efficiency gap between neural network model training and asynchronous processor design.

## 2 RELATED WORK

The term "asynchronous processing" is often used whenever neurons can be active in parallel and communicate asynchronously, even if some synchronization protocol is enforced to control the order of spikes and synaptic current processing Rathi et al. (2023); Kang et al. (2020b); Shahsavari et al. (2023); Yousefzadeh et al. (2019). Another kind of asynchrony is related to input coding Guo et al. (2021). In this context, synchronizing means grouping spikes (events) into frames, a topic that has extensively been researched He et al. (2020); Messikommer et al. (2020); Qiao et al. (2024); Ren et al. (2024); Taylor et al. (2024). Neither of the two, however, is the focus of this work. Here, asynchrony refers solely to activity within the network not being artificially bound by any order restriction. This has been researched for simulating biologically accurate neural networks Lytton & Hines (2005); Magalhães et al. (2019; 2020), but remains under-explored in the context of SNNs for machine learning. In this context, processing in layers is just one convenient way Mo & Tao (2022).

Few event-driven neuromorphic processors, such as μBrain Stuijt et al. (2021) and Speck Caccavella et al. (2023), are fully event-driven and lack any layer synchronization mechanism. This makes them notorious for training out-of-the-box with mainstream model training tools that rely on per-layer synchronization (e.g., PyTorch). Speck developers propose to train their models with hardware in-the-loop Liu et al. (2024) to reduce the mismatch between the training algorithms and the inference

hardware. However, this method does not provide a general solution for training asynchronous neural networks. Others, like SpiNNaker Furber et al. (2014) and TrueNorth Akopyan et al. (2015), use timer-based synchronization, or Loihi Davies et al. (2018) and Seneca Tang et al. (2023) use barrier synchronization between layers, in order to warrant that the asynchronous processing dynamics on hardware are aligned at layer boundaries with models trained in software (with layer synchronization). This, however, entails an efficiency penalty, as we will show.

Functionally, the most suitable type of model for end-to-end asynchronous processing is probably rate-coded SNN models, whereby neurons can integrate state and communicate independently from any other neuron. These models are trained like ANNs Zenke & Vogels (2021); Lin et al. (2018); Diehl et al. (2015) or converted from pre-trained ANNs Rueckauer et al. (2017); Kim et al. (2020). Therefore, to be accurate under asynchronous processing, it is required to run the inference for a long time, reducing the latency and energy benefit of using SNNs Sengupta et al. (2019).

Alternative to rate-coding models are temporal-coding models, with time-to-first spike (TTFS) Park et al. (2020); Srivatsa et al. (2020); Kheradpisheh & Masquelier (2020); Comşa et al. (2022) or order encoding Bonilla et al. (2022). They are very sparse (hence energy efficient) but very cumbersome and elaborate to convert from ANNs Painkras et al. (2013); Srivatsa et al. (2020) or train directly Kheradpisheh & Masquelier (2020); Comşa et al. (2022), less tolerant to noise Comşa et al. (2022), and their execution so far, while event-based, requires some form of synchronization or a reference time. Efficiency is thus only attributable to the reduced number of spikes, all of which need to be evaluated in order before a decision is reached. Other alternative encodings for event-based processing of SNNs include phase-coding Kim et al. (2018) and burst coding Park et al. (2019), which are, however, no more economical than rate coding and have not been shown, to our knowledge, to attain competitive performance.

The approach presented in this paper is, in fact, unique in enabling the trainability of models for event-based asynchronous execution, providing efficiency from processing only a subset of spikes, and delivering consistent performance and tolerance to noise. Also relevant to the works in this paper are SparseProp Engelken (2024) and EventProp Wunderlich & Pehle (2021) on efficient event-based simulation and training, respectively. EventProp Wunderlich & Pehle (2021) is potentially more economical than discrete-time backpropagation for training event-based models for asynchronous processing (and fully compatible with the work in this paper), but it has not been shown how its complexity scales beyond 1 hidden layer. SparseProp Engelken (2024) proposes an efficient neuron execution scheduling strategy for asynchronous processing in software simulation. An effective hardware implementation of this scheduling strategy can be found in Monti et al. (2017), which has inspired the herein proposed *"momentum schedule"*. Moreover, we advance this research by demonstrating how to train SNN models using this event scheduler.

## 3 METHODS FOR SIMULATING AND TRAINING OF ASYNCHRONOUS SNNS

In this section, we provide an overview of our methodology. We start by showing how one can simulate asynchronous SNN processing (implementing network asynchrony) and then we explain how this can be integrated in back-propagation training, to prepare good performing and efficient asynchronous models.

### 3.1 SIMULATING ASYNCHRONOUS SNNS

The SNNs used in this work consist of $L$ layers of Leaky Integrate-and-Fire (LIF) neurons He et al. (2020), with each layer $l$ for $1 \leq l \leq L$ having $N^{(l)}$ neurons and being fully-connected to the next layer $l+1$, except for the output layer. Each input feature is connected to all neurons in the first layer. Every connection has a synaptic weight. Using these weights, we can compute the incoming current $x$ per neuron resulting from spikes, as per equation 3.

Each LIF neuron has a membrane potential exponentially decaying over time based on some membrane time constant $\tau_m$. We use the analytical solution (see equation 4) to compute the decay. By keeping track of the membrane potential $u[t]$ and the elapsed time $\Delta t$ since time $t$, it is possible to precisely calculate the membrane potential for $t + \Delta t$. Therefore, computations are required only when $x[t + \Delta t] > 0$.

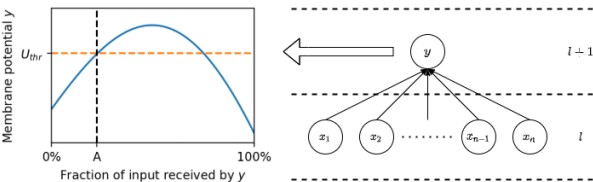

Figure 1: With network asynchrony, currents from neurons $x_i$ in layer $l$ may come to neuron $y$ in layer $l + 1$ at any time in any order. It may spike before all inputs have been received in a discrete timestep. Illustrated is the exceedance of the firing threshold at point $A$. All inputs, and thus information after $A$ is missed, i.e., the spiking decision is made based on partial information. In this example, if layer synchronization is enforced, neuron $y$ would wait for 100% of the input and will not spike.

To determine if a neuron spikes, a threshold function $\Theta$ checks if the membrane potential exceeds a threshold $U_{\text{thr}}$, following equation 5. If $\Theta(u) = 1$, the membrane potential is hard reset to 0. Soft reset is another option Guo et al. (2022) as well as refractoriness Sanaullah et al. (2023). However, for simplicity here we only experiment with hard reset.

### 3.1.1 EVENT-DRIVEN STATE UPDATES

We can vectorize the computations introduced in the previous section to perform them on a per-layer basis. This is herein referred to as the "layered" inference approach, which implies layer synchronization, since all neurons of one layer need to evaluate their state, before any neuron in a down-stream layer does the same. An alternative to this situation is an event-driven approach, where the computations for state updates are applied in response to new spike arrivals. If additionally spikes can be emitted (threshold evaluation) at any point in time (e.g in response to any individual synaptic current), by any neuron in the network, affecting the states of postsynaptic neurons independently of others, then (we assume) this approach can achieve a true representation of network asynchrony.

When using network asynchrony, the network dynamics evolve with each spike. Any single spike can generate multiple currents downstream, each linked to a specific stimuli at timestep $t$, determined by the initiating input activity. Let $K_t$ represent the atomic computation that is executed at one neuron in the network, in response to stimulus presented at the network at timestep t, and which will update the state of the neuron and evaluate its threshold function (activation). The exact computations for a LIF neuron can be found in section A.2. The order in which $K_t$ is executed across neurons inside the entire network (and not just in one layer) affects the output of the network due to the non-linearity in $K_t$ (see figure 1 for an example of the effect this can have). Here for simplicity, it is assumed that the fan-out currents resulting from the same spike are processed as a group at once (e.g. no heterogeneous propagation delays exist), and processing is done when an emitted spike is scheduled for processing, i.e., when the spike is propagated. This may not necessarily be the same moment as when the spike was emitted due to network asynchrony. Because of these assumptions, analyzing the spike propagation order gives the same insights as analyzing the execution order of $K_t$ (across neurons in the network).

Unlike the layered approach, in this modus operandi because of the per-input current evaluation of the firing threshold neurons could theoretically fire more than once at the same timestep $t$ (in response to the same input stimulus). We prevent this by inducing a neuron to enter a refractory state for the remainder of time $t$. In this state, the neuron keeps its membrane potential at 0. This makes the model causally simpler, more tractable, and more "economic" when energy consumption is coupled to spike communication. These aspects may also give explicit retrospective justification for the neurophysically observed refractoriness Berry & Meister (1997).

**Two resolutions of time** We apply input framing as typically done for inputs from a DVS sensor, accumulating timestamped events in discrete time-bins, and providing these frames as inputs to the network in discrete *timesteps*. The *timestep size* refers to the time-interval between the timestamps of the first (or last) events in the time-bins of two consecutive timesteps. Timesteps correspond to one concept of timing (resolution) with same semantics as in RNNs, and which we refer to here as

---

**Algorithm 1** Vectorized network asynchrony forward pass

---

**Input:** input spikes $\mathbf{s}_{\text{in}} \in \mathbb{N}^{N_{\text{in}}}$, previous forward pass time $t_0 \in \mathbb{R}$, current forward pass time $t_1 \in \mathbb{R}$, neuron state $\mathbf{u} \in \mathbb{R}^N$ at time $t_0$, forward group size $F \in \mathbb{N}_{>0}$
**Output:** spike count $\mathbf{c} \in \mathbb{N}^N$
   $\Delta t \leftarrow t_1 - t_0$
   $\mathbf{u} \leftarrow \text{NeuronDecay}(\mathbf{u}, \Delta t)$
   $\mathbf{x} \leftarrow \text{InputLayerForward}(\mathbf{s}_{\text{in}})$
   $\mathbf{s} \leftarrow \mathbf{0}^N$                                                ▷ Vector with zeros of length $N$
   $\mathbf{c} \leftarrow \mathbf{0}^N$
   **while** $(\text{Sum}(\mathbf{x}) \neq 0 \textbf{ or } \text{Sum}(\mathbf{s}) \neq 0)$ **and** $\neg\text{EarlyStop}(\mathbf{s})$ **do**
      $\mathbf{s}_{\text{new}}, \mathbf{u} \leftarrow \text{NeuronForward}(\mathbf{x}, \mathbf{u})$
      $\mathbf{s} \leftarrow \mathbf{s} + \mathbf{s}_{\text{new}}$                                     ▷ Enqueue new spikes
      $\mathbf{c} \leftarrow \mathbf{c} + \mathbf{s}_{\text{new}}$                                    ▷ Update spike count
      $\mathbf{s}_{\text{selected}} \leftarrow \text{SelectSpikes}(\mathbf{s}, F)$
      $\mathbf{s} \leftarrow \mathbf{s} - \mathbf{s}_{\text{selected}}$                               ▷ Dequeue selected spikes
      $\mathbf{x} \leftarrow \text{NetworkLayersForward}(\mathbf{s}_{\text{selected}})$
   **end while**

---

*macro-time*. At each timestep, input event frames initiate network activity, in a so-called *forward pass*, making all new spikes contingent on the timestep.

The order of spikes propagated though the network during a timestep establishes another notion of "time" in a forward pass (we call it "micro-time"), which is discretized in so-called *forward steps*. A forward step is associated with the processing of one or a group of events (in case of vectorization), and the number of forward steps measures time to complete inference. One can realize that in case of layered inference the number of forward steps is fixed and equals the number of layers, but this is not the case for asynchronous processing.

### 3.1.2 VECTORIZED NETWORK ASYNCHRONY

The event-driven (neuron state) update rules for network asynchrony as introduced in the previous section can be vectorized by selecting a number of spikes for processing them simultaneously. This allows us to consider the entire spectrum of possibilities between per-layer synchronization at one extreme (by assuming a vector size equal to a layer size), and "complete" asynchrony at the other extreme, where each spike event is processed entirely independently of all others. Additionally, vectorization makes acceleration possible by exploiting the parallelization features and vector pipelines of accelerators, where these models execute, leading to pragmatic simulation of network asynchrony.

During the simulation, the states of all the $N = \sum_{l=1}^{L} N^{(l)}$ neurons in the network are stored in vectors. Vector $\mathbf{x} \in \mathbb{R}^N$ tracks the computed input currents for the neurons, $\mathbf{u} \in \mathbb{R}^N$ the membrane potential of the neurons, $\mathbf{s} \in \mathbb{N}^N$ the emitted spikes awaiting processing, and $\mathbf{c} \in \mathbb{N}^N$ which neurons have spiked in the current forward pass. For any of those vectors, the indices from $\sum_{k=1}^{l-1} N^{(k)}$ to $\sum_{k=1}^{l} N^{(k)}$ represent the values for the neurons in layer $l$, for $1 \leq l \leq L$ where $N^{(0)} = 0$.

Algorithm 1 outlines the processing during a forward pass (propagation of the spikes of an input frame through the network). Input spikes are available in $\mathbf{s}_{\text{in}} \in \mathbb{N}^{N_{\text{in}}}$ (not to be confused with $\mathbf{s}$) where $N_{\text{in}}$ is the number of input features. Each *forward pass* consists of *forward steps* (the code within the while loop), which update the state of a set of neurons based on the spikes selected for propagation. A complete list of parameters can be found in section A.5.

The forward pass ends either when all spike activations have been processed ("On spiking done" stop condition), or when any neuron in the output layer has spiked one or more (default is one) forward steps ago ("On output" stop condition). The latter is checked in the EarlyStop function.

The SelectSpikes function defines how to select a subset of the emitted spikes for propagation. The function selects $F$ spikes at a time, or less if there are less than $F$ remaining spikes ready to be propagated. The method of spike selection is determined by a *scheduling policy*.

For the experiments here two policies are exercised. The intent is that different scheduling strategies can be tested for their effectiveness in capturing tractably asynchronous processing dynamics. The

first, Random Scheduling (RS), randomly picks spikes from the entire network. The second, Momentum Scheduling (MS), prioritizes spikes from neurons based on their membrane potential upon exceeding their threshold.

The neuron model-specific (LIF) behavior is expressed in the NeuronDecay and NeuronForward functions. This entails the computations for state updates (see section 3.1) for all neurons in the network (NeuronDecay) or for only those neurons receiving input currents in the forward step (NeuronForward), with added restriction that spiking is only allowed once per forward pass.

The network architecture is defined by the InputLayerForward and NetworkLayersForward functions. These functions compute the values of synaptic currents from spikes. This follows from equation 3.

**Imitating neuromorphic accelerator hardware**   Neuromorphic processors come in a number of variations, but most of them have a template architecture that interconnects many tiny processing cores with each other. This design enables a scalable architecture that supports fully distributed memory and compute systems. In this template, each processing core has its own small synchronous execution domain, but the cores operate asynchronously among each other. Architectures such as those cited in the Section 2 all follow this template. Our asynchronous processing simulation environment captures the intrinsics of the synchronous domain (e.g. in the forward group size $F$) while simulating the asynchronous interactions among them (e.g. scheduling policy) and can be also configured to reproduce more specialized intrinsic behaviors of many of those processors (table 5).

## 3.2 Training asynchronous SNNs

We use backpropagation to train the model weights, and specifically when stimulus is presented across multiple timesteps (forward passes), such as for sequential or temporal data, then this is Backpropagation Through Time (BPTT). Following common practice to address the non-differentiability of the threshold function, the surrogate gradient method is used Zenke & Vogels (2021). Specifically the arctan function Fang et al. (2021), (see section A.3 for details) provides a continuous and smooth approximation of the threshold function.

Class prediction is based on the softmaxed membrane potentials (for CIFAR-10) or spike counts over time (for the other datasets) of the neurons in the output layer, as described in section A.4. The loss is minimized using the Adam optimizer, with $\beta_1 = 0.9$ and $\beta_2 = 0.999$.

### 3.2.1 Unlayered backpropagation

We refer to "conventional" SNN training with per-layer synchronization using backpropagation Dampfhoffer et al. (2023), as "layered backpropagation".

The vectorized network asynchronous processing approach is differentiable as well, and can be used with backpropagation. We refer to this as "unlayered backpropagation". Combined with BPTT unrolling, this method implies a two level unrolling. At the outer level, unrolling is based on discretization of time and thus the input across (as usually fixed number of) timesteps. At the inner level, unrolling is based on the $F$-grouping (and vectorized processing) of spikes in forward steps, subject to the scheduling policy, applied to the emitted spikes by neurons; from the beginning of the current timestep until the output is read.

Note, that the dependence on the scheduling policy, applied on a variable number of activations (across timesteps and data samples), in $F$-sized groups is the fundamental difference from layered back-propagation, and leads to different gradient state being built up in the computation graph. This state now captures the dynamics of asynchronous processing. For a single *backward pass* in a timestep of BPTT, which is applied in a similar way as the layered backpropagation equivalent, it is given that:

$$\frac{\partial L_t}{\partial W} = \frac{\partial L_t}{\partial c_t} \sum_{i=1}^{N_t} \left( \frac{\partial c_t}{\partial s_i} \frac{\partial s_i}{\partial W} \right) \tag{1}$$

where $t$ is the time(step) of the *forward pass*, $W$ refers to the trainable weights, $L_t$ is the loss, $c_t$ is the spike count at the end of the forward pass, and $s_i$ is the emitted spikes vector at the end of the forward step $i$. The overall gradient (state) depends on the total number of *forward steps* $N_t$ in the forward pass and the spikes processed in each forward step. The number of steps scales linearly

with the number of spikes processed in the forward pass, and it is important to understand that due to asynchronous processing the number of spikes processed until the evaluation of the loss function may be (well) less than the total number of spikes emitted throughout the forward pass. Since for every forward step, the computations are repeated, the time complexity scales linearly with the number of forward steps, $O(N_t)$. The same applies to the space complexity.

During the backward pass, the spikes in $s_i$ which are not selected for processing can skip the computation $f$ for that step:

$$\frac{\partial s_i}{\partial W} = \frac{\partial f(s_{i-1;\text{selected}}, u_{i-1}, x_t, W)}{\partial W} + \frac{\partial s_{i-1;\text{not selected}}}{\partial W} \qquad (2)$$

Skip connections have been researched in deep ANNs and identified as a contributor to the stability of the training process Orhan & Pitkow (2018). This may apply to the skip in unlayered backpropagation as well. The extent to which this is the case remains unexplored in this work.

Note that under this generalization, layered backpropagation corresponds to $F$-group the size of a layer, populated with a static round-Robin execution schedule following neuron index order within a layer, and re-reset across consecutive layers).

### 3.2.2 REGULARIZATION TECHNIQUES

During training, we use regularization to prevent overfitting and/or enhance model generalization. These techniques are not used during inference.

**Input spike dropout**. Randomly omits input spikes with a given probability. The decision to drop each spike is independent according to a Bernoulli distribution.

**Weight regularization**. Adds weight decay to the loss function: $L_{\lambda_W}(\mathbf{W}) = L(\mathbf{W}) + \lambda_W \|\mathbf{W}\|_2^2$ where $\lambda_W$ is the regularization coefficient, $L$ is the loss given the weights, and $\mathbf{W}$ are all the weights.

**Refractory dropout**. With some probability, do not apply the refractory effect, allowing a neuron to fire again within the same forward pass.

**Momentum noise**. When using momentum scheduling, noise sampled from $U(0, 1)$ and multiplied by some constant $\lambda_{\text{MS}}$ is added to the recorded membrane potential while selecting spikes.

## 4 RESULTS

### 4.1 EXPERIMENTAL SETUP

We carried out our experiments on SNN models trained primarily in three common benchmarking datasets, each of them has a different structure: N-MNIST Orchard et al. (2015), SHD Cramer et al. (2020), and DVS gestures Amir et al. (2017). N-MNIST has purely spatial structure, SHD purely temporal, and DVS gestures combines both spatial and temporal (input framing in DVS gesture is done such that an entire gesture motion and contour is not revealed within a single frame). We also repeat some of the experiments with a fourth dataset, CIFAR-10 Krizhevsky (2012), in a more complex VGG-style network, which we will discuss in section 5. More details on the datasets and network topologies can be found in section A.6.1. Table 1 summarizes the parameterization of the experiments and the reported results. The network architecture and hyperparameters are given in table 6. State-of-the art performance for these tasks can be achieved with reasonably shallow and wide models. We chose however to train narrow, but deeper network architectures so that the effects of absence of layer synchronization can be revealed in the comparison (because of this bias in network topology the accuracy results shown can vary from the top state of the art). In section A.7, we also provide an additional ablation with regard to how various training hyperparameters affect accuracy (forward group size, refractory dropout, and momentum noise during training). Results on error convergence are provided in section A.9.

### 4.2 NETWORK ASYNCHRONY INCREASES NEURON REACTIVITY

As observed in figure 2 (top row) during asynchronous inference, neurons are more reactive, i.e. a neuron can spike after integrating only a small number of incoming currents. With layer synchronization, this effect is averaged out as neurons are always required to consider all presynaptic currents

Table 1: Parameterization of experiments and results.

| Parameterization | Description |
| --- | --- |
| Training method | Training with layered backpropagation is marked as "Layered" and with unlayered backpropagation as "Unlayered [scheduling strategy]". |
| Inference method | Inference with layer synchronization is marked as "Layered" and with network asynchrony as "Async [scheduling strategy]". |
| Scheduling strategy | How to select spikes from the queue. Can be random ("RS"), or based on the membrane potential just before spiking ("MS"). |
| Forward group size $F$ | Number of spikes to select for processing at the same time. Default is 8, both during training and inference. |
| Stop condition | Forward pass terminates: If all network activity drains ("On spiking done") or one forward step after the first spike is emitted by the output layer ("On output"). The default is "On spiking done" during training and "On output" during inference. |

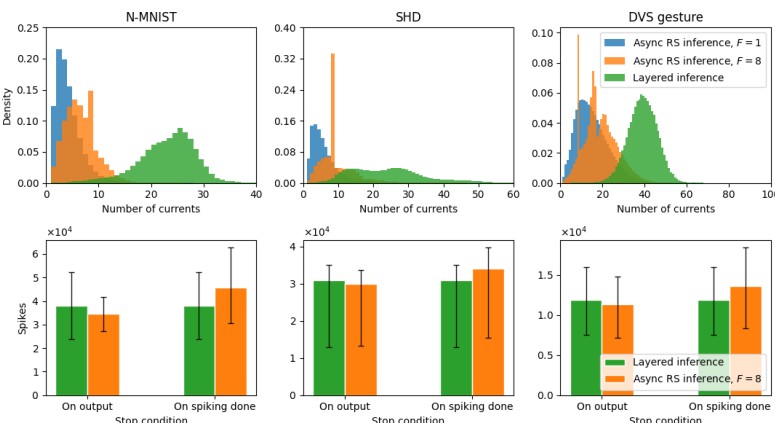

Figure 2: **(Top row)** Number of currents integrated by a neuron before spiking, recorded per neuron and per forward pass for all samples and neurons (excluding the neurons in the input layer). The Y-axis shows the relative frequency of the number of currents integrated before spiking. **(Bottom row)** Mean number of spikes per neuron during inference of all samples in the test. Error bars show the 25th and 75th percentiles. (Models in this figure were trained with layered backpropagation.)

(which are more likely to cancel each other out). For inference with $F = 8$, artifacts in the number of currents to spike can be observed, because each forward step propagates currents resulting from 8 spikes, causing neurons to integrate more currents than necessary for firing. Similar artifacts might also occur in neuromorphic chips equipped with fixed-width vector processing pipelines; making these observations insightful into the behavior of such hardware.

One expects that more reactive neurons imply higher activation density. Interestingly, this is not necessarily the case for the models trained for asynchronous inference! In figure 2 (bottom row) we see that if we wait for the network to "drain" of spike activity during a forward pass the total number of spikes will indeed be higher. But if the forward pass terminates as soon as a decision is made, asynchronous models are consistently sparser. This is because asynchronous processing allows spike activity to freely flow through to the output and not be blocked at every layer for synchronization.

### 4.3 UNLAYERED BACKPROPAGATION RECOVERS ACCURACY AND INCREASES SPARSITY

Network asynchrony negatively affects the performance of the models trained with layered backpropagation in all three datasets (table 2). This is likely the "Achilles' heel" of neuromorphic processing today. However, we observe that the accuracy loss is remediated when training takes into account asynchronous processing dynamics, which also significantly increases sparsity (by about 2x). We witnessed that the accuracy of models trained and executed asynchronously is consistently superior under the two scheduling policies we considered. This result conjectures that neuromorphic AI is competitive and more computationally efficient.

Table 2: Accuracy and activation density results. More details about these metrics in section A.6.2.

| Training | Inference | N-MNIST | | SHD | | DVS gesture | |
| | | Acc. | Density | Acc. | Density | Acc. | Density |
|---|---|---|---|---|---|---|---|
| Layered | Layered | 0.949 | 3.987 | 0.783 | 14.386 | 0.739 | 46.473 |
| Layered | Async RS | 0.625 | 3.652 | 0.750 | 13.905 | 0.701 | 44.140 |
| Unlayered RS | Async RS | 0.956 | 1.504 | 0.796 | **5.100** | 0.777 | **25.140** |
| Unlayered MS | Async MS | **0.963** | **1.476** | **0.816** | 6.224 | **0.856** | 26.686 |

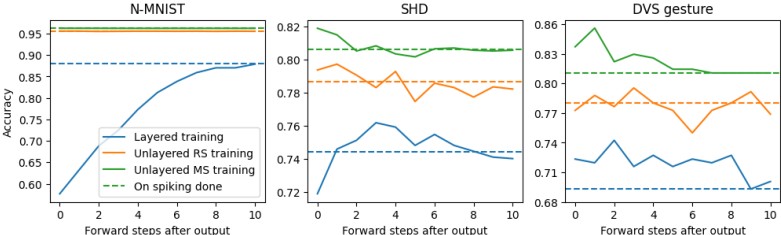

Figure 3: Accuracy as function of forward steps after the first spike in the output layer. Given that $F = 8$, each extra forward step processes another 8 spikes, assuming enough spikes are available. Dashed lines show the accuracy after all spike activity has been "drained" out of the network.

Figure 3 reveals another interesting result. It depicts how accuracy evolves as we allow more forward steps in the forward pass after the initial output during asynchronous inference. We see that because of the free flow of key information *depth-first*, models trained with unlayered backpropagation obtain the correct predictions as soon as the output layer gets stimulated. Activity that is likely triggered by "noise" in the input is integrated later on. Momentum scheduling is particularly good at exploiting this to boost accuracy.

## 4.4 NETWORK ASYNCHRONY AND ENERGY EFFICIENCY

To quantify the energy savings from asynchronous processing in the trained models, table 3 presents the mean number of synaptic operations and energy consumption on the μBrain neuromorphic chip Stuijt et al. (2021) per sample. This accounts for dynamic power savings. More details also on static power savings in a digital (neuromorphic) accelerator are provided in section A.11.

Table 3: Mean num of Synaptic Operations (SOs $\times 10^4$) and energy consumption (μJ) for classification of one sample, using 26 pJ/SO as measured for μBrain Stuijt et al. (2021); ignoring static power.

| Training | Inference | N-MNIST | | SHD | | DVS gesture | |
| | | SOs | Energy | SOs | Energy | SOs | Energy |
|---|---|---|---|---|---|---|---|
| Layered | Layered | 3.521 | 0.9155 | 50.82 | 13.21 | 158.8 | 41.29 |
| Layered | Async RS | 3.225 | 0.8386 | 49.12 | 12.77 | 150.9 | 39.22 |
| Unlayered RS | Async RS | 1.328 | 0.3454 | 18.02 | 4.684 | 85.92 | 22.34 |
| Unlayered MS | Async MS | 1.304 | 0.3389 | 21.99 | 5.717 | 91.2 | 23.71 |

## 4.5 NETWORK ASYNCHRONY REDUCES LATENCY

An equally important result concerns the inference latency reduction under asynchronous network processing. The models trained with unlayered backpropagation have significantly lower latency than those trained with layer synchronization. Assuming a unit latency for processing one spike, the comparatively worst case latency will be given under sequential processing, as a function of the number of spikes processed until an inference decision is made. Figure 4 shows a distribution of the inference latencies across the entire test-set. It should be clear by now that this is because "the important spikes" in these models quickly reach the output layer, uninhibited by layer synchronization.

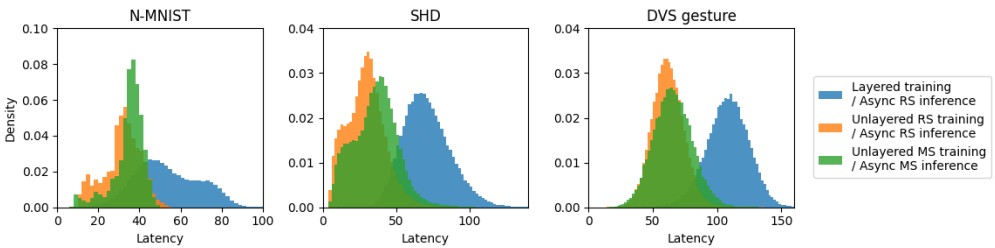

Figure 4: Latency per forward pass for all samples, in number of spikes until a decision in the output layer is reached. The Y-axis shows the relative frequency of recorded latencies.

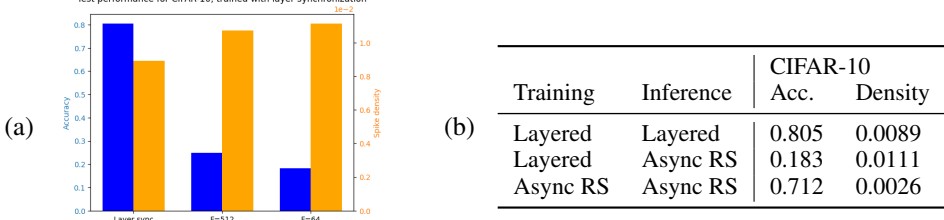

(a)     (b)

| Training | Inference | CIFAR-10 Acc. | Density |
|----------|-----------|---------------|---------|
| Layered | Layered | 0.805 | 0.0089 |
| Layered | Async RS | 0.183 | 0.0111 |
| Async RS | Async RS | 0.712 | 0.0026 |

Figure 5: (a) Impact of network asynchrony on a CIFAR-10 trained VGG-style model. A larger decrease in accuracy occurs with smaller forward group sizes accompanied by an increase in activations. (b) Accuracy and activation density results on CIFAR-10.

### 4.6 UNLAYERED BACKPROPAGATION IS RESOURCE-INTENSIVE

We also tried to confirm the results in a scaled up setup, namely with a deeper VGG-7 like network (details in A.10), trained on the CIFAR-10 dataset. Figure 5a confirms the problem when training with layered backpropagation and the running asynchronous inference. When we try to train asynchronous models with unlayered backpropagation and the simpler RS scheduling policy the memory cost however becomes prohibitive unless we substantially increase the forward group size $F$. Nevertheless, even with as large as $F = 512$ (during training, the smallest possible with an NVIDIA RTX 5000 GPU) and tested with $F = 64$, accuracy steeply recovers to 71%, while activation density drops to about $1/4$ (confirming our previous observations). We anticipate that with much smaller $F$ during training (higher degree of network asynchrony), the accuracy can be completely recovered.

Unfortunately, the computational cost of unlayered backpropagation, in the current framework (and implementation) is rather high, especially when training with a small $F$. Each time $F$ is halved, the time and memory requirements approximately double as discussed in sections 3.2.1 and A.8.

## 5 DISCUSSION & FUTURE OUTLOOK

We pinpoint a crux for neuromorphic AI processing in training SNN models conventionally with backpropagation and naively assuming they will execute consistently, and energy and latency efficiently in neuromorphic processors. The problem lies in neglecting the dynamics of asynchronous processing, and we found a way to factor them in the gradient training process. Resulting models not only recover the affected performance (even exceed it) but also exhibit the energy and latency benefits touted by neuromorphic computing. Execution scheduling strategies may be a missing bridge between neural algorithms/models and neuromorphic hardware architecting. This study merely scratched the surface of a research exploration in this direction, but it shows that unless we rethink our training methods to account for asynchronous dynamics and redesign neuromorphic accelerators to move away from layer synchronization primitives, SNNs and brain-inspired computing could fail to delivering both performance and efficiency. Our future work needs to tackle challenges in 2 directions: improving on the computational limitations of our asynchronous training approach (and affirming our results in larger/deeper models), and bridging this work to more complex and more contemporary model structures (e.g. Deng et al. (2022); Yao et al. (2022); Guo et al. (2023b); Yu et al. (2022)).

## REPRODUCIBILITY STATEMENT

The methods and appendices should provide enough information to reproduce the results. To help with the reproducibility, we also provided code.

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

# A    Appendix / supplemental material

## A.1    Spiking neural networks

### A.1.1    Incoming current

For every neuron layer, all the synaptic weights on its inbound connections are kept in a weight matrix $\mathbf{W}^{(l)} \in \mathbb{R}^{N^{(l)} \times N^{(l-1)}}$, where $N^{(0)} =$ number of input features $N_{\text{in}}$. Using these weight matrices, the total incoming current $x$ for a neuron $i$ in the next layer can be computed using:

$$x_i^{(l+1)}[t] = \sum_{j=1}^{N^{(l)}} W_{ij}^{(l+1)} s_j^{(l)}[t] \tag{3}$$

where $s_j^{(l)}[t]$ is 1 if neuron $j$ in layer $l$ has emitted a spike at time $t$, otherwise 0.

### A.1.2    Membrane potential decay

The decay of the membrane potential is governed by a linear Ordinary Differential Equation (ODE). The analytical solution can be used to compute the decay:

$$u[t] = u[t - \Delta t] \cdot e^{-\frac{\Delta t}{\tau_m}} + x[t] \tag{4}$$

where $\tau_m$ is the membrane time constant and $\Delta$t the elapsed time.

### A.1.3 THRESHOLD FUNCTION

$$\Theta(u) = \begin{cases} 1 & \text{if } u > U_{\text{thr}} \\ 0 & \text{otherwise} \end{cases} \tag{5}$$

where $U_{\text{thr}}$ is the membrane potential threshold required for spiking.

### A.2 EVENT-DRIVEN STATE UPDATE RULE FOR LIF NEURON

When an input current is received at time $t$ by a neuron with a previous state $u[t_0]$ at some time $t_0 \le t$, an atomic set of computations is executed. Start with computing the decayed membrane potential $u[t_-] = u[t_0] \cdot e^{\frac{-(t-t_0)}{\tau_m}}$, then update the membrane potential $u[t_+] = u[t_-] + x[t]$ with the input current $x[t]$, and finally set $u[t] = 0$ and emit a spike if $\Theta(u[t_+]) = 1$; otherwise $u[t] = u[t_+]$ without emitting a spike.

### A.3 ARCTAN SURROGATE GRADIENT

$$\Theta(u) = \frac{1}{\pi}\arctan(\pi u \frac{\alpha}{2}) \tag{6}$$

where $\alpha$ is a hyperparameter modifying the steepness of the function.

### A.4 CLASS PREDICTION

Class prediction involves first calculating the output rates as follows:

$$c_i = \sum_{t \in T} s[i, t] \tag{7}$$

where $c_i$ is the spike count for class $i$, $T$ are all the timesteps for which a forward pass occurred, and $s[i, t]$ is the output of the neuron representing class $i$ in the output layer at the end of the forward pass at time $t$. For CIFAR-10, instead the value of $c_i$ is equal to the membrane potential of the output neuron for class $i$ at the end of having processed all the spikes. These values are subsequently used as logits within a softmax function:

$$p_i = \frac{e^{c_i}}{\sum_{j=1}^{N_C} e^{c_j}} \tag{8}$$

where $N_C$ is the total number of classes. The resulting probabilities are then used to compute the cross-entropy loss:

$$L = \sum_{i=1}^{N_C} y_i \log(p_i) \tag{9}$$

where $y \in \{0, 1\}^{N_C}$ is the target class in a one-hot encoded format.

## A.5 PARAMETERS FOR THE SIMULATOR

Table 4: Overview of all current simulator parameters. If the text is *italic*, then the parameter was not used for the experiments in this paper.

| Name | Value range | Description |
|------|-------------|-------------|
| Forward group size | $\mathbb{N}_{>0}$ | 3.1.2 |
| Scheduling policy | RS/MS | 3.1.2 |
| Prioritize input | True/False | If input spikes are propagated before any spikes from inside the network. |
| Stop condition | On spiking done / On output | 3.1.2 |
| Forward steps after output | $\mathbb{N}_{\geq 0}$ | 3.1.2 (only for "On output" stop condition) |
| Refractory dropout | $[0.0, 1.0]$ | 3.2.2 |
| Momentum noise | $\mathbb{R}_{\geq 0}$ | 3.1.2 (only for "MS" scheduling policy) |
| Membrane time constant | $\mathbb{R}_{>0}$ | 3.1 |
| Input spike dropout | $[0.0, 1.0]$ | 3.2.2 |
| *Network spike dropout* | $[0.0, 1.0]$ | The same as input spike dropout, but for spikes from inside the network, applied per forward step. |
| Membrane potential threshold | $\mathbb{R}_{>0}$ | 3.1 |
| Timestep size | $\mathbb{N}_{>0}$ | 3.1.1 |
| *Synchronization threshold* | $\mathbb{N}_{\geq 0}$ | Have neurons wait for a number of input currents (including barrier messages) before being allowed to fire. Can be set per neuron. |
| *Emit barrier messages* | True/False | Have neurons emit a barrier message if they have retrieved exactly the number of input currents to exceed the synchronization threshold, but not enough to exceed the membrane potential threshold. |

Table 5: Simulator parameters for simulating neuromorphic processors

| Synchronization type | Neuromorphic processors | Forward group size | Synchronization threshold [emit barrier messages] |
|------|------|------|------|
| Barrier-based / layer | Loihi Davies et al. (2018), SENECA Yousefzadeh et al. (2022), POETS Shahsavari et al. (2021) | # neu/layer | # neu/layer [True] |
| Timer-based / core | SENECA Yousefzadeh et al. (2022), SpiNNaker Painkras et al. (2013), TrueNorth Akopyan et al. (2015), POETS Shahsavari et al. (2021) | # neu/core | # neu/core [True] |
| Asynchronous | Speck Caccavella et al. (2023), µBrian Stuijt et al. (2021) | # neu/layer | 1 [False] |

## A.6 DETAILS ON THE EXPERIMENTAL SETUP

### A.6.1 DATASETS AND NETWORK ARCHITECTURES

The Neuromorphic MNIST (N-MNIST) dataset captures the MNIST digits using a Dynamic Vision Sensor (DVS) camera. It presents minimal temporal structure Iyer et al. (2021). It consists of 60000

training samples and 10000 test samples. Each sample spans approximately 300 ms, divided into three 100 ms camera sweeps over the same digit. Only the initial 100 ms segment of each sample is used in this study.

The Spiking Heidelberg Digits (SHD) dataset Cramer et al. (2020) is composed of auditory recordings with significant temporal structure. It consists of 8156 training samples and 2264 test samples. Each sample includes recordings of 20 spoken digits transformed into spike sequences using a cochlear model, capturing the rich dynamics of auditory processing. The 700 cochlear model output channels are downsampled to 350 channels.

The DVS gesture dataset Amir et al. (2017) focuses on different hand and arm gestures recorded by a DVS camera. Like the SHD dataset, it has significant temporal structure. It focuses on 11 different hand and arm gestures recorded by a DVS camera. It consists of 1176 training samples and 288 test samples. The $128 \times 128$ input frame is downsampled to a $32 \times 32$ frame.

CIFAR-10 Krizhevsky (2012) is a widely-used image classification dataset consisting of 60000 32x32 color images across 10 classes, including animals and vehicles. It has no temporal structure. Unlike the other datasets, all input is provided in one single timestep.

Table 6: Network architecture and hyperparameters. The architecture is given as [neurons in hidden layers $\times$ number of hidden layers] - [neurons in output layer].

|  | N-MNIST | SHD | DVS gesture | CIFAR-10 |
|---|---|---|---|---|
| Architecture | [64×3]-10 | [128×3]-20 | [128×3]-11 | see A.10 |
| Timestep size | 10 ms | 10 ms | 20 ms | N/A |
| Batch size | 256 | 32 | 32 | 64 |
| Epochs | 50 | 100 | 70 | 150 |
| Learning rate | 5e-4 | 7e-4 | 1e-4 | 2e-4 |
| Membrane threshold $U_{\text{thr}}$ | 0.3 | 0.3 | 0.3 | 0.2 |
| Weight decay constant $\lambda_W$ | 1e-5 | 1e-4 | 1e-5 | 0 |
| Membrane time constant $\tau_m$ | 1 ms | 100 ms | 100 ms | N/A |
| Surrogate steepness $\alpha$ | 2 | 10 | 10 | 2 |
| Input spike dropout | 0.25 | 0.2 | 0.2 | 0 |
| Forward group size $F$ | 8 | 8 | 8 | 512 |
| Refractory dropout | 0.8 | 0.8 | 0.8 | 0.7 |
| Momentum noise $\lambda_{\text{MS}}$ | 1e-6 | 0.1 | 0.1 | see A.10 |

For all four datasets, events of a data point belong to a continuous stream where they have a timestamp and an index position (see figure 6). In the case of N-MNIST, the index corresponds to a position within a $34 \times 34$ pixel frame, with each pixel having a binary polarity value (either 1 or 0), leading to a total of $34 \times 34 \times 2 = 2312$ distinct input indices. For SHD, the index denotes one of the 350 frequency channels. For DVS gesture, the $128 \times 128$ input frame is downsampled to a $32 \times 32$ frame. Like N-MNIST, each pixel has a binary polarity value, so in total this gives $32 \times 32 \times 2 = 2048$ input indices. Unlike the other datasets, for CIFAR-10, the input events present a continuous value (so they are currents instead of spikes), while also still being assigned an index and timestamp (although the timestamp is irrelevant). For encoding them as inputs to the models the stream of events is sliced in time-bins $(\epsilon_k, \epsilon_{k+1}, ...)$ along the temporal dimension and the timestamps are discretized to the time-bin index. The time-bins then construct time-frames whereby all the events at each spatial index are accumulated (counted). Each time-frame is then used as input to the model at discrete subsequent timesteps. This is one of the common-place input encoding for SNNs.

### A.6.2 PERFORMANCE METRICS

To evaluate accuracy, output rates $c_i$ for each class $i$ are first calculated as outlined in equation 7. The predicted class corresponds to the one with the highest output rate. Accuracy is then quantified as the ratio of correctly predicted outputs to the total number of samples.

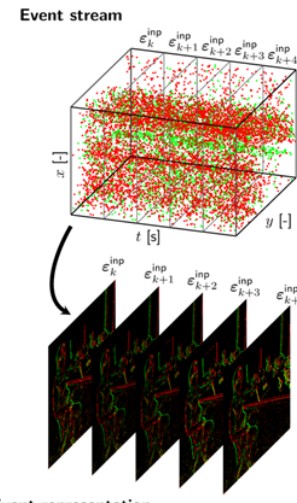

Figure 6: Encoding of a continuous temporal stream of input events into frames Hagenaars et al. (2021).

Spike density is computed using:

$$\text{density} = \frac{1}{N_{\text{samples}} \cdot N_{\text{neurons}}} \sum_{i=1}^{N_{\text{samples}}} N_{\text{spikes}}[i] \tag{10}$$

where $N_{\text{samples}}$ is the number of samples, $N_{\text{neurons}}$ is the number of neurons in the hidden layers, and $N_{\text{spikes}}[i]$ is the total number of spikes during inference of the sample $i$.

## A.7 RESULTS ON HYPERPARAMETERS

Choosing a smaller $F$ (i.e., with a more asynchronous system), may improve accuracy, particularly benefiting models with mechanisms that rely on network asynchrony such as momentum scheduling. However, reducing $F$ also has its drawbacks. It significantly raises resource demands (discussed in section A.8), and there is a risk of reducing the effectiveness or even stalling the training process, as observed for SHD and DVS gesture.

Refractory dropout, can positively affect training outcomes. An explanation for this is that it increases the gradient flow by allowing more spiking activity. However, using full refractory dropout can also reduce performance, likely due to the inability to generalize to inference with refractoriness.

The momentum noise helps by introducing a slight stochastic element into the spike selection process, helping to avoid potential local minima that a purely deterministic selection method is prone to get stuck in. This seems to do little for N-MNIST, but for more complex datasets like SHD and DVS gesture it has a significant effect.

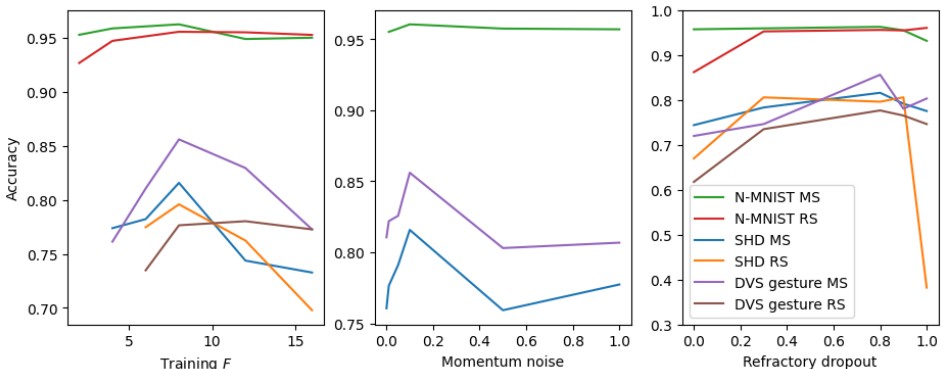

Figure 7: Results from inference with network asynchrony for different hyperparameters used during training. For SHD and DVS gesture with random scheduling, training $F$s smaller than 6 are not included due to the training failing to converge.

## A.8 RESULTS ON RESOURCE USAGE

The increase in processing time is less pronounced for the N-MNIST and DVS gesture datasets compared to the SHD dataset. This discrepancy could be due to computational optimizations that apply specifically to the N-MNIST and DVS gesture datasets (both being vision-based datasets).

Table 7: Resource usage compared between layered and unlayered backpropagation during the second epoch of training on an NVIDIA Quadro RTX 5000.

| Method | Time per epoch (s) | VRAM use (MB) |
|---|---|---|
| **N-MNIST** | | |
| Layered | 14 | 210 |
| Unlayered RS, training $F = 16$ | 19 | 412 |
| Unlayered RS, training $F = 8$ | 24 | 556 |
| Unlayered RS, training $F = 4$ | 37 | 986 |
| **SHD** | | |
| Layered | 38 | 214 |
| Unlayered RS, training $F = 16$ | 118 | 2220 |
| Unlayered RS, training $F = 8$ | 292 | 4844 |
| Unlayered RS, training $F = 4$ | 681 | 10574 |
| **DVS gesture** | | |
| Layered | 120 | 244 |
| Unlayered RS, training $F = 16$ | 150 | 3802 |
| Unlayered RS, training $F = 8$ | 177 | 6984 |
| Unlayered RS, training $F = 4$ | 245 | 13914 |

## A.9 RESULTS ON ERROR CONVERGENCE

Both unlayered training with random scheduling and momentum scheduling achieve lower loss values compared to layered training, as shown in figure 8. At the start of training, random scheduling shows a slightly less steep loss curve, while momentum scheduling shows a steepness comparable to that of layered training. For CIFAR-10, error convergence is shown in figure 10.

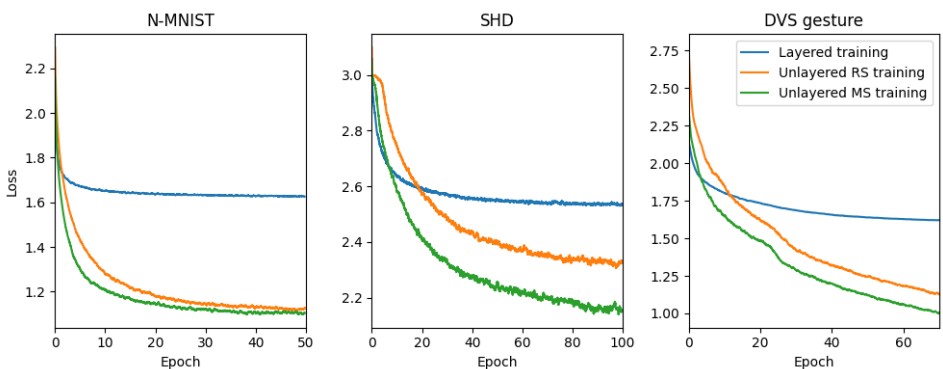

Figure 8: Error convergence for different synchronization methods.

## A.10 DETAILS ON CIFAR-10 DATASET WITH VGG EXPERIMENTS

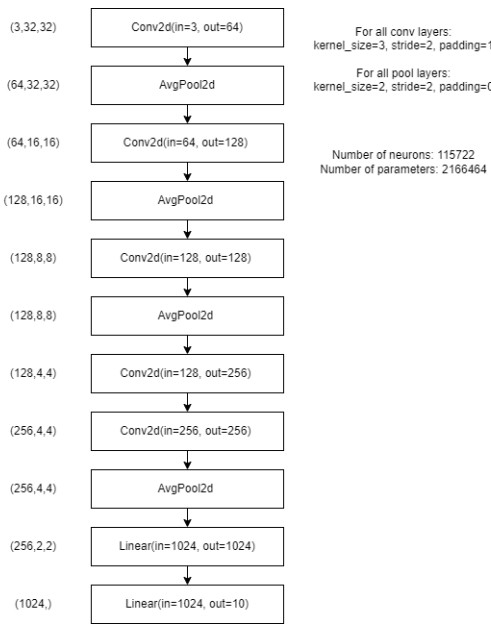

Figure 9: Simplified VGG network topology without batch normalization and dropout layers.

The details of network structure of the VGG model that we used to train on the CIFAR-10 dataset is shown in figure 9. We have removed (for simplicity) the batch normalization and dropout layers.

To reduce the memory overhead for training with unlayered backpropagation and make it feasible on an NVIDIA RTX 5000 we provided all the input of each sample in a single timestep (thus eliminating the unfolding across timesteps), and we adopted a forward group size $F = 512$ for training and $F = 64$ for testing. Additionally we deployed Integrate-and-Fire (IF) neurons that do not leak.

Output predictions were based on the membrane potential accumulated at the output neurons. Because of the very large $F$, MS scheduling policy was difficult to train requiring rather large amounts of annealing noise (the range of values tested are shown in figure 10).

The error convergence can be found in figure 10. Unlike the other datasets, layered training converges towards a lower loss value and with a slightly steeper curve than the unlayered methods. For momentum scheduling with $\lambda_{MS} = 0.1$ or $\lambda_{MS} = 2.5$, training was stopped early due to failure to convergence.

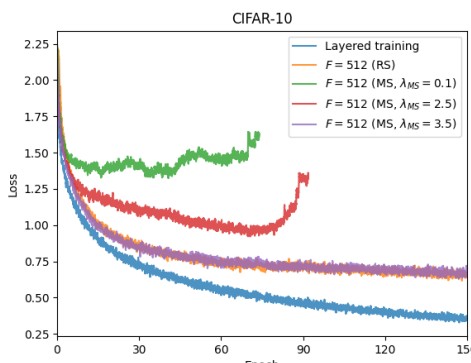

Figure 10: Error convergence for CIFAR-10 for different synchronization methods and amounts of momentum noise in the MS scheduling policy.

## A.11 NEUROMORPHIC PROCESSING AND ENERGY EFFICIENCY

Neuromorphic chips such as µBrain Stuijt et al. (2021) and Speck Caccavella et al. (2023) alike are at the forefront of asynchronous spiking neural network (SNN) hardware development, showcasing significant benefits over traditional clock-based architectures. Both chips employ an event-driven, fully asynchronous, architecture, eliminating the need for global or per-layer synchronization. In both cases, all neurons can fire a spike immediately once their membrane potential crosses a specific threshold in response to integrating an incoming current/spike (without waiting for a synchronization signal or an interval timer to expire). Such fully event-driven inference allows independent and instantaneous neuron processing, reducing computational overhead and latency, and directly matches the operational principles of SNNs. However, neurons with fully event-driven asynchronous computation are sensitive to the sub-microsecond timing and exact order dynamics of the incoming spikes, which a time-stepped training algorithm is not sensitive to. As a result, there will be a significant accuracy drop if the deployed SNN of these chips is trained with a time-stepped algorithm.

Traditional synchronous simulation methods for SNNs introduce limitations when mapping to these asynchronous neuromorphic chips. In synchronous simulations, time is discretized into uniform steps, and all neurons in a layer are updated simultaneously, leading to excessive idle computations and artificial delays that are not representative of "real-time" temporal dynamics. In contrast, asynchronous simulation methods align more closely with the intrinsic event-driven nature of neuromorphic hardware. They allow all neurons to react as events occur, mirroring the operational principle of chips like µBrain and Speck. This results in lower latency and more efficient mapping of SNN models onto the hardware.

In the absence of common-place asynchronous training and simulation methods, many other neuromorphic chips (e.g. SpiNNaker Furber et al. (2014), TrueNorth Akopyan et al. (2015), Loihi Davies et al. (2018), Seneca Tang et al. (2023)), resort to explicit layer synchronization primitives (timer-based current integration or explicit signaling) for maintaining consistent performance with synchronous simulated and trained models. Under this constraint these systems execute models event-based, but not end-to-end asynchronously (even though they could). This comes at the cost of increased latency of execution and energy consumption.

Typically energy consumption on such digital neuromorphic accelerators has two components. A baseline static one that is present even when the system is idle so long as it is powered, and which relates to its hardware attributes and components (memory leakage, clock frequency, manufacturing technology, and other). And a dynamic one that relates to the operation of the system for executing a model. The latter has to do with actual computations the system performs, memory IO (typically the source of the Von Neumann bottleneck), and communication (particularly in multicore systems the entail a network-on-chip). In many architectures a technique called power-gating helps save substantial energy from static power by switching off parts of the system that are idle.

Model execution is directly related to energy consumption due to dynamic power (e.g. when fetching data from memory and executing arithmetic operations), but also indirectly due to static power (as a function of the inference latency that forces the system to stay powered).

