Source code of the work in the paper paper can be found here: `https://anonymous.4open.science/r/AsyncInf-ICLR`

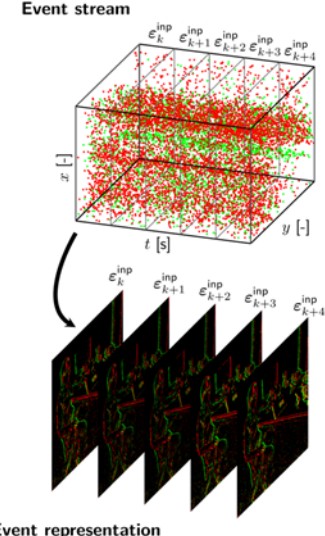

Figure 1: Discretization of a continuous temporal stream of external input events, and encoding into frames. Source of the image is [1].

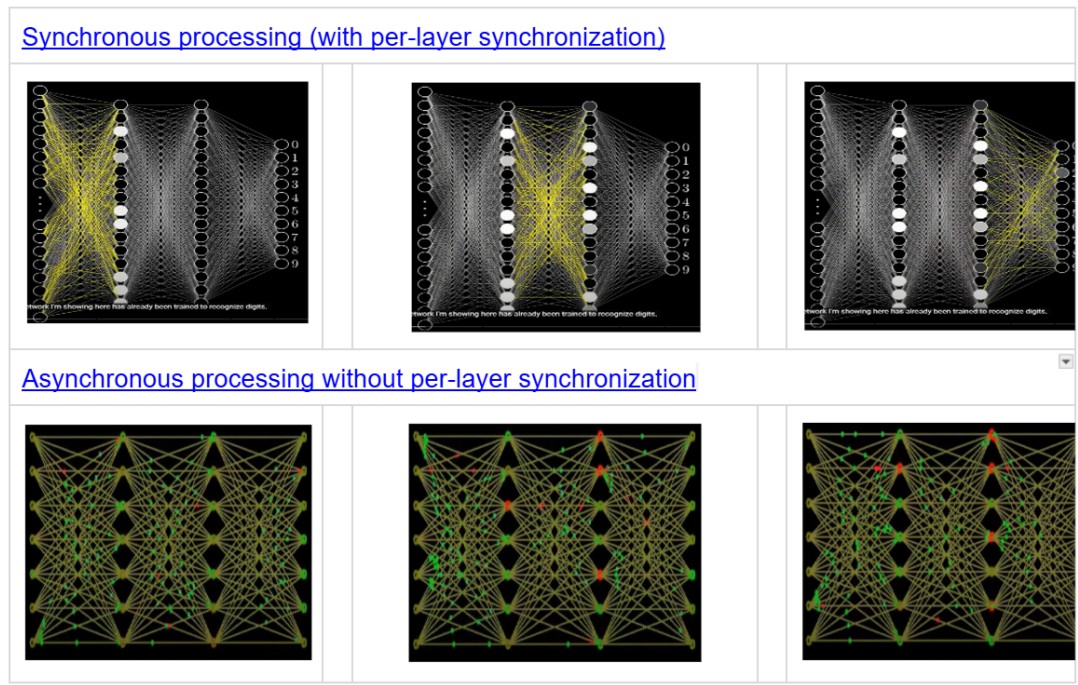

Figure 2: Layer synchronized processing compared to asynchronous processing. These are snapshots from full animations that can be found here: `https://anonymous.4open.science/r/AsyncInf-ICLR/Sync_inference.mp4` for layer synchronized processing, and `https://anonymous.4open.science/r/AsyncInf-ICLR/Async_inference.mp4` for asynchronous processing.

.

# References

[1]  Jesse Hagenaars, Federico Paredes-Vallés, and Guido de Croon. "Self-Supervised Learning of Event-Based Optical Flow with Spiking Neural Networks". In: *Advances in Neural Information Processing Systems* 34 (2021).