# OpenReview forum: "Exploring the Limitations of Layer Synchronization in Spiking Neural Networks"
_ICLR.cc/2025/Conference — Submitted to ICLR 2025_

### Official Review · Reviewer_Vpkp · 2024-10-22

**Soundness:** 3
**Presentation:** 2
**Contribution:** 2
**Rating:** 3
**Confidence:** 3

**Summary:**

This paper proposes an asynchronous simulation method for deep SNNs. The network structure, vectorized simulation algorithm, backward methods, and experiment results on some simple datasets are provided.

**Strengths:**

As we know, most (or all?) existing software frameworks for deep SNNs use a (layer-wise) synchronous simulation method. The idea of the asynchronous simulation in this paper is very interesting. The source codes in the supplementary materials indicate that the authors have developed an asynchronous primary software framework, which will benefit the SNN community.

**Weaknesses:**

Unfortunately, this paper is written unclearly. The difference between synchronous and asynchronous simulation methods is not explained clearly. It is better to illustrate the difference by a figure, i.e., an example to show thow the synchronous/asynchronous method simulates an SNN.


The advantages of using the asynchronous simulation method are not shown. I guess that this simulation method is more similar to how the asynchronous neuromorphic chip works. However, I believe that not all of SNN researchers are familiar to the hardware, and it is hard for them to understand the advantages of asynchronous simulation without the knowledge about neuromorphic chips I suggest that the authors add more background knowledge about asynchronous neuromorphic chips (such as uBrain and Speck), and why the existing simulation methods are not so good for them such as the difference between synchronous simluation and asynchronous on-chip inference.

**Questions:**

I am not sure that I really understand the methods in this paper. Thus, I have the following questions. Please feel free to clarify my wrong understanding.

What is the role of `c` in Algorithm 1? It is only used as the output.

I am not sure if the key characteristic of the proposed asynchronous method is "selecting spikes". More specifically, suppose a layer has `N` inputs, the previous synchronous simulation method uses the spike tensor `S` with `N` elements as inputs. The proposed method will firstly apply a scheduling strategy to create a `mask` to select elements (spikes) in `S`, and use `S[mask]` as inputs. If so, it seems that this characteristic is trying to mimic the asynchronous spike firing in chips, which do not have a global clock. The role of the global clock is similar to "time-step" in SNNs simulated by the synchronous method. Then, what is the relation between the asynchronous spike firing in chips and the randomly selecting in simulation? I guess (note that I am not the expert of neuromorphic hardware) that the events (spikes) are processed (chronologically) orderly in chips, and this behavior is more similar to the synchronous simulation with a large number of time-steps. While the randomly selecting is disordered.

---

> ### Author Response · Authors · 2024-11-21
> **Answer to question(s) relevant to the difference between synchronous and asynchronous simulation**
>
> NB. references to sections and line numbers cited in the following answers refer to the original version of the manuscript.
>
> In order to provide more coherent explanations below we tried to dissect and group together the review comments which raise similar points and that could be addressed in the same answer. We kindly ask the reviewer to suggest if/which of these answers are essential to include in the paper or supplementary material for improving the clarity and quality of our work.
>
> > The difference between synchronous and asynchronous simulation methods is not explained clearly. It is better to illustrate the difference by a figure, i.e., an example to show how the synchronous/asynchronous method simulates an SNN.
>
> > … why the existing simulation methods are not so good for them such as the difference between synchronous simluation and asynchronous on-chip inference
>
> We agree that 1 photo == 1000 words
>
> In figure 2 in the updated supplementary material we provide snapshots from 2 videos (from the internet) that visualise the fundamental difference between synchronous processing with per-layer synchronisation and asynchronous processing. If you think it is necessary for improving the quality and understanding of the paper, we can add parts of the following discussion in the paper appendix.
>
> As one can see in the provided figures, in the case of layer-synchronous execution, spikes from one layer are not propagated to the next unless processing of all neurons in the presynaptic layer is completed. Timing or order information of spikes across input synapses in each neuron is lost since all currents are integrated altogether before the membrane threshold is evaluated (only once per integration interval). The dynamics of spikes are therefore nulled and execution follows “breadth first order” (all neurons from one layer get evaluated before neurons of the next layer start their evaluation). This creates all sorts of issues which we enumerate in the answer of the followup question.
>
> By contrast, in the case of asynchronous processing without layer-synchronisation spikes arriving from any synapse at any neuron in the network are integrated immediately triggering the membrane threshold evaluation independently of any other (section 3.1.1) . In absence of layer synchronisation (barriers) they propagate further downstream before other neurons in the same layer complete their evaluation. This makes the flow dynamics of activations to follow the relative timing/order of spikes, making any part of the network potentially active at any moment in time and allowing “depth first order” of execution if the flow dynamics require it. This is the type of in-network operation that our simulation environment supports particularly when parameter `F=1`, and what neuromorphic accelerators support (analog ones, and digital ones like μBrain and Speck as cited in the paper)
>
> When it comes to digital accelerators (because they discretise processing) they often may employ small degrees of vectorization (`F>1` in our simulation environment), to batch event processing and reduce memory I/O. This means that for a small number of spikes they may collapse timing/order and align them to process them in batch. This tampers/interferes with timing/ordering of spikes at a very small scale, but since it does it is not restricted only to spikes/currents delivered to (adjacent) neurons of the same layer, it does not have the adverse effects and extent of enforcing per-layer synchronisation, so as to hamper the activation dynamics. It can merely be perceived as (annealing) noise, and therefore asynchronous processing still works. Our simulation and training framework takes these aspects into account to armourplate the model against these effects and preserve/recover the good accuracy.

---

> > ### Author Response · Authors · 2024-11-21
> > **Answer about clarity of the advantages of asynchronous processing**
> >
> > > advantages of using the asynchronous simulation method are not shown.
> >
> > Following from the respective videos and illustrations added in the supplementary materials (see previous question), we hope it is easy to affirm the following
> >
> > The system executing synchronously with per-layer synchronisation performs either layer computations or inter-layer communication, in successive phases, where the two are mutually exclusive. This creates a situation where there are memory IO bottlenecks (to fetch state), followed by computation bottlenecks, followed by memory IO bottlenecks (to update state in memory), followed by communication bottlenecks (to propagate events), and this repeats all over again at every synchronisation point. Furthermore in this modus operandi all activation traffic (spikes) needs to be exhaustively processed before a decision at the output layer is made. This type of processing “buys” us tractability and coherence between a software trained model and inference executed on accelerator hardware at the expense of the aforementioned overheads (that cost energy), and latency.
> >
> > By contrast, in the system executing asynchronously (and event-driven), computation and communication are interleaved, avoiding all the bottlenecks discussed above. In absence of synchronisation points, activations propagate fast through the entire network, reducing latency, and inference can terminate prematurely before all activations are exhaustively processed, leading to sparse computations (which also reduce the memory IO). Any additional sparsity in the activations is also exploited to further avoid unnecessary delays and computations. This buys us energy efficiency and latency reduction. This is what brain-inspired/neuromorphic computing is assumed to be about (according to neuroscience). In addition latency reduction means that the uptime of the system is reduced, which additionally saves on static power in the cases of digital circuits that leak (such as DRAM) and thus further push down energy consumption. But the skeleton in the closet in this case is that models executed in this way are not coherent with synchronous training (accuracy loss) and often/traditionally intractable … unless we find a different way to train them for such modus operandi.
> >
> > Our work is essentially addressing this crux. By capturing critical aspects of asynchronous processing (from neuromorphic accelerators) in the simulation environment, as highlighted in the answer to the previous question, and accounting for these aspects in training (essentially abstracting them in BP training to facilitate asynchronous training), we demonstrate that it is possible to capitalise on asynchronous inference in all three fronts: accuracy, energy, latency.

---

> > > ### Author Response · Authors · 2024-11-22
> > > **Answer to the request for more background info in neuromorphic chips**
> > >
> > > > I guess that this simulation method is more similar to how the asynchronous neuromorphic chip works. However, I believe that not all of SNN researchers are familiar to the hardware, and it is hard for them to understand the advantages of asynchronous simulation without the knowledge about neuromorphic chips I suggest that the authors add more background knowledge about asynchronous neuromorphic chips (such as uBrain and Speck),
> > >
> > > In addition to the explanations provided to the previous answers we will added a section in the appendix with some more details about these neuromorpchic accelerators and about dataflow event-based accelerators in general, based on the following decription and the relevant aspects touched upon in the answers to the other reviewers too.
> > >
> > > Neuromorphic chips such as µBrain and Speck alike are at the forefront of asynchronous spiking neural network (SNN) hardware development, showcasing significant benefits over traditional clock-based architectures. Both chips employ a fully event-driven (asynchronous) architecture, eliminating the need for global or per-layer synchronisation. In both cases, all neurons can fire a spike immediately once their membrane potential crosses a specific threshold in response to integrating an incoming current/spike (without waiting for a synchronisation signal or an interval timer to expire). Such fully event-driven inference allows independent and instantaneous neuron processing, reducing computational overhead and latency, and directly matches the operational principles of SNNs. However, neurons with fully event-driven asynchronous computation are sensitive to the sub-microsecond timing and exact order dynamics of the incoming spikes, which a time-stepped training algorithm is not sensitive to. As a result, there will be a significant accuracy drop if the deployed SNN of these chips is trained with a time-stepped algorithm.
> > >
> > > Traditional synchronous simulation methods for SNNs introduce limitations when mapping to these asynchronous neuromorphic chips. In synchronous simulations, time is discretized into uniform steps, and all neurons in a layer are updated simultaneously, leading to excessive idle computations and artificial delays that are not representative of “real-time” temporal dynamics. In contrast, asynchronous simulation methods align more closely with the intrinsic event-driven nature of neuromorphic hardware. They allow all neurons to react as events occur, mirroring the operational principle of chips like µBrain and Speck. This results in lower latency and more efficient mapping of SNN models onto the hardware.

---

> > > > ### Author Response · Authors · 2024-11-22
> > > > **Asnwers to the requests for clarifications to the Algorithm**
> > > >
> > > > > role of c in Algo
> > > >
> > > > In Algorithm 1, c is an output vector that keeps track of the neurons that have fired spikes and their count across all forward steps, i.e., it is used to track when and how often neurons fire in the entire network. This is important for classification and loss calculation, statistics, and for establishing termination of inference conditions. For example, for enforcing the “On output condition” we need to know when (at which forward step) output neurons start firing. In such cases, simply tracking which neurons spike during a single forward step (vector s) is insufficient.
> > > >
> > > > > … key characteristic of the proposed asynchronous method is "selecting spikes".
> > > >
> > > > > If so, it seems that this characteristic is trying to mimic the asynchronous spike firing in chips, which do not have a global clock
> > > >
> > > > > The role of the global clock is similar to "time-step" in SNNs simulated by the synchronous method.
> > > >
> > > > > what is the relation between the asynchronous spike firing in chips and the randomly selecting in simulation?
> > > >
> > > > > I guess (note that I am not the expert of neuromorphic hardware) that the events (spikes) are processed (chronologically) orderly in chips, and this behavior is more similar to the synchronous simulation with a large number of time-steps. While the randomly selecting is disordered.
> > > >
> > > > We answer each part of this multi-question in distinct paragraphs below
> > > >
> > > > **One of the key characteristics** of the proposed asynchronous method is “selecting spikes” (and scheduling them for execution), **but it is not the only one**. As mentioned in a previous answer **we allow neurons to integrate immediately incoming currents and evaluate the firing threshold for every current**. This preserves the stimulus dynamics at each neuron’s instantaneous membrane. We also **allow spike activity to propagate through network layers without synchronisation** (a neuron at the output layer can fire before a neuron at the first hidden layer has responded to stimulus). This is **facilitated through adaptive execution scheduling**, which thus preserves the activation propagation dynamics among neurons and layers (i.e. selecting spikes anywhere in the network). A scheduling policy also reflects **neuron evaluation intrinsics about various neuromorphic hardware accelerators**. And we also capture **aspects of vectorization of digital dataflow (neuromorphic) accelerators** that can influence the asynchronous processing dynamics. Finally and most importantly we abstract *all* these aspects inside the model training, where we optimise for accuracy performance.
> > > >
> > > > These characteristics together try to mimic not only the asynchronous spike firing in chips, but also the **asynchronous processing end-to-end** ! (we remind: no layer-synchronisation, vectorisation, order of evaluation, and instantaneous flow dynamics preservation in neurons).
> > > >
> > > > With all respect, we disagree with the statement that “””the global clock is similar to "time-step" in SNNs simulated by the synchronous method”””. The **time-stepping in SNNs/RNNs only clocks the sequential admission of external input to the network. It has nothing to do with what is happening inside the network thereafter between timesteps**. In the synchronous processing case it is however prescribed and tractable what happens (that is the effect or per-layer synchronisation – paid at a high energy/latency cost as we explained in a previous answer). In an asynchronous processing method nothing is prescribed, and so to abide to synchronous-trained models in many neuromorphic accelerators extra primitives for synchronisation are often offered/enforced (explicit signals or timers). When using these primitives then processing is only event-driven but not asynchronous end-to-end !
> > > >
> > > > Finally the case of **Random Scheduling, reflects the fact that the spike propagation and neuron integration process is or can be inherently noisy (either epistemically or aleatorically), thus changing the temporal/rank-order of spikes**, irrespective of their exact timing of occurence. It basically can be seen as annealing noise or dropout noise, which beyond a certain level breaks the system down but in small enough quantities (relevant to the vector pipelining of the accelerator – `F-grouping` in our framework), makes the system more robust. But note that it is applied across the entire network not layer-after-layer! (so that it does not block the spike propagation dynamics).
> > > >
> > > > With this last set of explanations we hope that we have managed to shed some better light to the contributions and quality of our work.

---

> > > > > ### Author Response · Authors · 2024-11-26
> > > > >
> > > > > We think that we have fully addressed your concerns with our explanations and updated pdf. We hope that you can update your rating if you are satisfied. Or alternatively pls ask us further questions or clarifications within the 1 day discussion period remaining.

---

> > > > > > ### Author Response · Authors · 2024-12-03
> > > > > >
> > > > > > We have not seen any follow-up or reaction or comments whatsoever to the inputs we have provided so far (and which would allow us to defend our work against your rating). In face of the deadline expiring very soon, we would like to kind ask you for any further comments, which we may try to address/clarify in the day remaining for the end of the rebuttal in order to improve our score.

---

### Official Review · Reviewer_LDn4 · 2024-11-02

**Soundness:** 3
**Presentation:** 2
**Contribution:** 2
**Rating:** 6
**Confidence:** 3

**Summary:**

This paper points out most current works train SNNs in a synchronous way, while SNNs should be running in an asynchronous environment, which leads to a gap. Then this paper quantifies this gap and develops a training method better suited for asynchronous inference. Experiments show the effectiveness of this method in accuracy, inference speed, and energy consumption.

**Strengths:**

This paper considers a critical problem for SNNs that they are expected to run in an asynchronous way, while most current SNN training strategies train them synchronously.

**Weaknesses:**

1. See questions.
2. There are some word and grammar mistakes and some figures should be improved. For example, 'compex' in line 363 should be 'complex', 'To what extent this is the case is not explored in this work.' in line 334 is incoherent, the legends in Figure 4 stretches over two subfigures.

**Questions:**

1. I am not sure about Algorithm 1. Does $\boldsymbol s$ represent whether there is a spike or the timing of the spike? If it represents the timing, why it is an integer? If it represents whether there is a spike, how to determine the arrival order of input spikes, which is critical in asynchronous simulation?
2. In SelectSpikes() in Algorithm 1, the two scheduling policies seem not to consider the difference in input spike arrival times. How the asynchornization is achieved then?
3. Due to the above reasons, I am not sure whether Algorithm 1 reflects the real asynchronous property of SNNs. If not so, the comparisons in the experiments are not fair since layered training with async RS inference is not a practical situation.
4. How the classification result is determined by the network? What do 'on output', 'on spiking done', and 'forward steps after output' in Figure 2 and Figure 3 mean?

---

> ### Author Response · Authors · 2024-11-21
> **Answer to first comment**
>
> NB. references to sections and line numbers cited in the following answers refer to the original version of the manuscript.
>
> Would like to thank the reviewer for the remarks and the opportunity to defend our work, make it better understandable, and improve its quality.
>
> > Does s represent whether there is a spike or the timing of the spike? If it represents the timing, why it is an integer? If it represents whether there is a spike, how to determine the arrival order of input spikes, which is critical in asynchronous simulation?
>
> In Algorithm 1, `s` is an int vector whose dimension equals the number of neurons in the network `N`, and conveys the information of which neurons in the network have fired/spiked (activation state in the network at any time point). Likewise `s_in` is analogous to `s` but for the input features (so it only has the dimension of the input layer `N_in`). The state in `s` is transitory across forward steps, and so it varies continuously. Thus the occurrence/presence of the spikes in it follow the discrete timing/order dynamics between forward steps. The concept of forward step is explained in lines 210-221, and again in 238-242 (but it is also further clarified in the answer of the follow up question raised later).
>
> To further clarify the misconception/ambiguity, in a (pure) event-based dataflow system there is **no hard requirement for explicit timestamping for asynchronous operation** in absence of explicit synchronisation points that stall the event propagation (e.g. at layers boundaries). Thus, events are propagated as they occur (based on flow dynamics). In our simulation environment (and in many neuromorphic accelerators) the **preservation of flow dynamics** is captured in the fact that **each incoming spike (current) in a neuron is immediately integrated and triggers validation of the membrane threshold** and potential firing (what we refer to as depth-first execution through the network).
>
> In practical reality digital accelerators may choose to use some degree of vectorization for batch event processing when events occur close to each other (which forces some time-alignment of these events). This architectural choice is captured in the concept of **forward groups** `F` in our algorithm. While this optimization tampers a bit with the “purity” of being single-event-driven it is there because it saves memory IO (which is important for digital platforms). Asynchrony nevertheless still works because this batching is not restricted to adjacent neurons in a layer only, and the forced time alignment/collapsing that occasionally emerges from it can be seen as (annealing) noise which the asynchronous system can learn to tolerate (far less severe than whole-layer synchronisation that blocks the flow dynamics).
>
> Another further point to keep in mind is that according to neuroscience theories (see literature cited in the citation [1] of reviewer FREL), exact timing of spikes appears to serve primarily the purpose of ordering. That is, the more critical information is captured in the ranking of spikes rather than the exact times, which justifies the emergence of rank-order and N-of-M codings as temporal/latency codes, and the attention to only the first spike in TTFS codes. So, again in this respect keeping track of timestamps is not critical so long as the enabled flow dynamics on average respect the ordering.

---

> ### Author Response · Authors · 2024-11-21
> **Answer to second and third comment**
>
> > In SelectSpikes() in Algorithm 1, the two scheduling policies seem not to consider the difference in input spike arrival times. How the asynchornization is achieved then?
>
> > Due to the above reasons, I am not sure whether Algorithm 1 reflects the real asynchronous property of SNNs. If not so, the comparisons in the experiments are not fair since layered training with async RS inference is not a practical situation.
>
> At a high level this may appear a reasonable concern, but we suspect the ambiguity arises in overlooking some of the details that provide the necessary information incrementally in various parts of the paper. Therefore below we try to bring all this information together and disambiguate the concern. If you find it essential for the understanding of the paper to have the following information in one place summarised, we can provide a dedicated section in the paper appendix.
>
> As we remarked in lines 114-116, and 172-175 **asynchronous processing is mainly about neurons acting independently of other neurons (also across layer boundaries) based on locally emerging dynamics**, which is feasible with rate coding. **So long as these dynamics are unobstructed by synchronisation barriers there is no need for explicit time-tracking**. Temporal codings which imply detailed time tracking may need explicit timestamps but this is also because of the presence of synchronisation points. Also, order codings and TTFS schemes may use timestamps to enforce strict per/across-layer ordering but they can also do without by working with queues. The overarching assumption, which is also supported by neuroscience, is that **exact timing is relevant mainly to the extent that it facilitates the relative ordering of events** (see previous answer).
>
> From this viewpoint, in our framework (sec 3.1.1) **synaptic currents arriving at a neuron are integrated immediately and each of them triggers evaluation of the membrane threshold independently of other incoming currents** (i.e. there is no integration time interval at the neuron level). This eliminates the need of keeping an explicit global clock for the spike times. The instantaneous membrane voltage of neurons then reflects the temporal dynamics (lines 176-188) of the incoming activations at the neuron level. The **dynamic scheduling allows for the order dynamics between neurons to be also reflected in the exchange of spikes without synchronization delays at layer boundaries**. In other words these mechanisms reflect relative temporal (order) dynamics among neurons across layers in the entire network. All this is accounting for asynchrony inside the network biassed only **(a)** from stimulus, and **(b)** from the way an event-based accelerator schedules event execution/processing (lines 277-287).
>
> Coming to the two scheduling policies we presented, they aim to represent these biases (lines 266-269), and integrate them in the model training (section 3.2.1).
>
> **Random Scheduling, reflects the fact that the spike propagation and neuron integration process is or can be inherently noisy (either epistemically or aleatorically), thus affecting the temporal/rank-order of spike occurrence**. It can be seen as annealing noise or dropout noise, which beyond a certain level breaks the system down but in small enough quantities (relevant to the vector pipelining of the accelerator), makes the system more robust. Note that it is applied across the entire network not layer-after-layer!
>
> **Momentum Scheduling, emphasis on the relative order dynamics in neuron evaluation as reflected in their membranes** (i.e. which neuron is likely to fire first next – relevance to TTFS and rank-order codes for the importance of the first spike(s) even though we work with rate codes). Note again this takes place across all layers, and remains unbiased by artificial layer-synchronisation barriers.
>
> The two scheduling policies tackle two different aspect of order dynamics. All this is meaningful primarily for neuromorphic and dataflow accelerators (for now). While we showcase only two scheduling policies, this is not to say these are the only possible ones. It is a topic of open exploration for the future, and co-design for dataflow AI accelerators.

---

> ### Author Response · Authors · 2024-11-21
> **Answer to fourth comment**
>
> > How the classification result is determined by the network? What do 'on output', 'on spiking done', and 'forward steps after output' in Figure 2 and Figure 3 mean?
>
> The stop conditions/criteria for inference are explained in lines 259-261 and in Table 1. `On spiking done` means that the network is left to drain of spike activity after presentation of stimulus (for a timestep), while `On output` means that spike processing terminates as soon as the output layer neurons start firing (which in absence of layer synchronisation can be very early before even all spike activity in the network has unfolded.
>
> The notion of `forward steps` and `forward pass` are explained in lines 210-221, and again in 238-242, and thereafter used in various parts of the text. In the temporal dimension, a network unrolls/unfolds across timesteps of input stimulus presentation (as standard for RNNs/SNNs and sequence models). At each timestep the network “unrolls” or unfolds spatially completely, layer-after-layer with synchronisation in-between. **In asynchronous processing the spatial unfolding is only in-parts**, and the parts that do they may follow any order, dictated by activation dynamics. To describe this partial unfolding of computations we use the term forward pass and within it forward steps, which corresponds to the number of computation steps (across the entire network) executed until a decision is made. The number of forward steps is thus typically variable, and depends on activation dynamics (spike activity integration speed and how it is propagated – i.e. the scheduling policy) as well as some accelerator characteristics (e.g. batching or vectorization primitives – `F` param in our algorithm).

---

> > ### Author Response · Authors · 2024-11-26
> >
> > We think that we have fully addressed your concerns with our explanations and updated pdf. We hope that you can update your rating if you are satisfied. Or alternatively pls ask us further questions or clarifications within the 1 day discussion period remaining.

---

> ### Comment · Reviewer_LDn4 · 2024-11-27
>
> Sorry for the late reply. I was busy with other things in the past few days.
>
> I still have some questions about SNN asynchronisation. If I have understood correctly, your core point about asynchrony is that, although you use synchronization techniques at last, it is easy to turn it into an asynchronized setting.
>
> However, in my opinion, the inference stage of common SNNs also easily turns into a synchronized setting (you can refer to Figure 1 in [1]), if you don't use modules like LayerNorm (BatchNorm in inference will not cause synchronizing problems). The core difficulty of asynchronisation is in training. The unlayered BP in 3.2.1 seems similar to BPTT, and I don't know what features related to asynchronization you have (you also discretize in training). Or do you have some features like only BP through spikes as in [1]? Could the authors emphasize this?
>
>
>
> > [1] Zhu, Y., Yu, Z., Fang, W., Xie, X., Huang, T., & Masquelier, T. (2022). Training spiking neural networks with event-driven backpropagation. *Advances in Neural Information Processing Systems*, *35*, 30528-30541.

---

> ### Author Response · Authors · 2024-11-27
>
> > I still have some questions about SNN asynchronisation. If I have understood correctly, your core point about asynchrony is that, although you use synchronization techniques at last, it is easy to turn it into an asynchronized setting.
>
> I think that in providing an coherent explanation to your question, first we need to make sure that we understand the same things when we refer to **event-based** and **asynchronous** processing, and the difference between **timesteps** at the model level and **processing steps** at the system level.
>
> By **event-based** we understand that a spike in an SNN (or non-zero activation in a DNN) can trigger independently some actions, individually and locally (that may or may not have a global effect). Event-based manifests essentially by comparison to vector-based. Because going to memory and back for every spike (for accessing state and weights) in **digital** neuromorphic accelerators is very expensive, by design they may choose to group spikes and process them together. So they may not be singe-event-based but may be 2-event or 4-event or N-event based (F-group in our formalization). NOTE this grouping of spikes is not restricted to spikes from the same neuron, or from neurons in the same layer as with vector accelerators in DNNs.
>
> By **asynchronous** we understand the fact that event responses can happen anytime (or in any order), and also anywhere in the network, independently of any other events AND unhindered by conditions that block their immediate propagation dynamics (such as waiting for all other neurons in the same layer to integrate their currents or evaluated their thresholds). Typically for a large system such as a network of neurons only approximate timing of events plays a role for asynchrony, and therefore order is more critical than exact timing (at least according to neuroscience literature). Approximate timing, makes asynchronous processing viable in digital accelerators even though they discretize time.
>
> **Timesteps** at the model level (and in [1], also pointed in our related work section) refer to the clocking, or discretization of time, or simply the order, that external stimulus is provided to the model (network). From the model perspective, between two such timesteps everything in the network is assumed to happen "atomically" or instantly, and the unrolling of traffic spatially from input to output is just a deterministic processing sequence, that falls out of the temporal realm of timesteps. There is an element of (very coarse grained) asynchrony in this context as to which spikes in the network, emerge at which timestep. Let us call this **"model asynchrony"** for now. This is what [1] is about, and it is the only aspect of asynchrony (model based) that is being looked at in the literature so far in our knowledge. But this is not what our paper is about.
>
> In this paper we look exactly into the spatial unrolling of traffic throughout the network between two consecutive timesteps of input stimulus. A neuromorphic system (and the brain for that matter) that processes one timestep's worth of stimulus can (and should) also operate asynchronously at this "spatial dimension" (for energy and latency reasons). Let us call this **"processing system asynchrony"**. This means that the evaluation order may not be strictly sequential and should not be dictated by the layer structure and the position of neurons in the network (only), but instead by spike dynamics (this is what the role of a scheduling policy in the paper is about), and potentially influenced by system features (such as whether the system processing is single-event-based or say 8-event-based). The latter gives rise to the concept of **forward processing steps** in the paper, as a finer-grained resolution of time (more correct order of spike evaluation) between timesteps of external stimuli at the model level.
>
> Essentially until now model-asynchrony and processing-system-asynchrony have not been connected or never looked at together (the former is what modelling researchers look at, but the latter is what neuromorpchic engineers are building!). In order to bridge the chasm, *layer-synchronisation* is used (in training and inference), even in event-based accelerators, which means that neurons of one layer are ALL evaluated (for their input spikes) before any neuron in the next layer gets evaluated. But this kills any dynamics due to processing-system-asynchrony because
>   - asynchrony is only limited within the scope of one layer in this way, and so
>   - an event-based and a vector-based processor will behave exactly the same (from the point of view of dynamics, and also often cost)
>
> In the paper we show the processing system asynchrony is what saves energy and latency, and also the bad things that happen (in performance) if layer synchronization is removed for a model not trained for executing asynchronously in the spatial dimension.

---

> ### Author Response · Authors · 2024-11-27
>
> > However, in my opinion, the inference stage of common SNNs also easily turns into a synchronized setting (you can refer to Figure 1 in [1]), if you don't use modules like LayerNorm (BatchNorm in inference will not cause synchronizing problems).
>
> Yes indeed, and that is the purpose that layer-synchronisation serves. But it comes at the expense of processing unnecessary traffic and interacting excessively with the memory in digital accelerators (Von Neumann bottleneck), consuming more energy, dissipating more heat, and incurring higher latency (practically compromising all the flagship benefits of neuromorphic computing). This is precisely the point of our paper. In practice we show that the difficult thing is the opposite, namely to turn a SNN that was trained for synchronous execution into an asynchronous one as it should be.
>
>
> > The core difficulty of asynchronisation is in training. The unlayered BP in 3.2.1 seems similar to BPTT, and I don't know what features related to asynchronization you have (you also discretize in training). Or do you have some features like only BP through spikes as in [1]? Could the authors emphasize this?
>
> So here is where I suspect we loose each other. You are referring to model-asynchrony, while we work on enabling processing-system-asychrony and connecting it to model-asynchrony (if you allow me the use of this custom terminology defined in the above part of the answer).
>
> Asynchronous learning updates with local rules (like STDP) is trivial overall. Asynchronous learning updates by applying BP asynchronously is (reasonably) difficult because of its global nature (but considered in various literature of optimization theory and learning theory in ANNs). However asynchronous processing (inference) and applying model updates asynchronously (training) are indeed two orthogonal things. Using synchronous learning updates with BP to train an asynchronous executing SNN, is the topic of this paper and with all respect we do not think is trivial (at least we have not seen this addressed with acceptable performance anywhere beyond 1-hidden layer networks, or demonstrated with any energy/latency advantages).
>
> **Unlayered BP is not about** trying to shift spikes around between timesteps using spike gating across timesteps in the backward pass (as in [1]) aiming to address **model asynchrony**. Instead it tries to **bridge model asynchrony with processing system asynchrony** through **in-training heuristics**. It effects that through
>   - preserving the instantaneous flow dynamics at the neuron level (each input current triggers the threshold evaluation independently -- Fig 1, lines 191-203)
>   - casting away layer-synchronization in forward and backward passes (section 3.1.1),
>   - embedding dynamic scheduling of neuron evaluation in the forward pass that leads to a entirely different and more dynamic across timesteps compute graph structure - and forward state (section 3.1.2)
>   - accounting in-training for vectorization and event batching features of neuromorphic processors (e.g Forward grouping - section 3.1.2, and keeping track of residue events instead of throwing them away).

---

> > ### Author Response · Authors · 2024-11-27
> >
> > Sorry for the lengthy response. We understand it is a bit difficult topic to digest between compute architectures and neural modelling but hope we addressed your question. We re available for any more clarifications.

---

> > > ### Comment · Reviewer_LDn4 · 2024-11-28
> > >
> > > My major concern is addressed. I have raised my score to 6.
> > >
> > > I hope these clarifications on concepts can be added to the paper (a clear illustrative figure is highly encouraged). Besides, it would be better to add a discussion on the relation and gap between Async RS / Async MS and real neuromorphic hardware.

---

> ### Author Response · Authors · 2024-11-30
>
> We'd like to thank you for this decision.
>
> We prepared a revision with two new added sections with these clarifications, but unfortunately as of 27/11 we cannot upload any more revisions. If we are permitted updates for a camera ready, we will provided them then.

---

### Official Review · Reviewer_z8jo · 2024-11-02

**Soundness:** 3
**Presentation:** 3
**Contribution:** 3
**Rating:** 8
**Confidence:** 3

**Summary:**

The paper highlights an important aspect of SNN, that is asynchronous computation of spikes, resulting in energy efficiency.  This work points out that, the training of SNN uses GPU and when deployed on asynchronous neuromorphic processes, the performance may be reduced during inference if asynchronous computation is adopted.

An generalized backpropagation algorithm is introduced which exploits the vectorized computation of the hardware and also allows asynchronous computation during inference without compromising the accuracy.

**Strengths:**

The paper is well-written and easy to follow.

Highlighted an important aspect of training SNNs (asynchronous computation) which is generally ignored while training SNN models.

The related work is very well explained, with their contributions and limitations.

Through empirical results, the paper demonstrates the effectiveness of the proposed algorithm in terms of sparsity and accuracy.

**Weaknesses:**

It's commendable that the authors have highlighted the asynchronous aspect of SNNs. Could the authors provide some comparative empirical results in terms of accuracy and sparsity with some previous works?

For example, the state of the art directly trained SNN models. Such as, [1,2,3], I believe they belong to the class of synchronous computation of spikes.

[1] Temporal Efficient Training of Spiking Neural Network via Gradient Re-weighting

[2] GLIF: A Unified Gated Leaky Integrate-and-Fire Neuron for Spiking Neural Networks

[3] Membrane Potential Batch Normalization for Spiking Neural Networks

**Questions:**

Check the Weaknesses.

---

> ### Author Response · Authors · 2024-11-21
> **part 1 of the answer**
>
> NB. references to sections and line numbers cited in the following answers refer to the original version of the manuscript.
>
> > comparative empirical results in terms of accuracy and sparsity with some previous works? For example, the state of the art directly trained SNN models. Such as, [1,2,3], I believe they belong to the class of synchronous computation of spikes.
>
> We appreciate the positive feedback and score, and thank the reviewer for raising the opportunity to better position our work in relation to results reported in the literature.
>
> We have read the suggested papers but as they involve experiments with larger datasets and far larger or more complex models we are not in position at the moment to carry out sound comparative experiments. Both because of time (complexity involved) and computational limitations (our training framework as is, reaches fast the limits of memory resources on our currently two GPUs for simulation – this is why we were limited with the VGG11 experiments in the first place).
>
> The suggested papers (as well as [1] from the first reviewer) help us pin-down specific directions for follow-up research however when it comes to scaling up and tackling more complex models. Specifically, making asynchronous processing viable in presence of batch-norm, auto-regression with attention, and gating mechanisms are all very interesting followup project work that cannot be concluded in the short time of the rebuttal. In parallel, addressing the computational cost of unlayered backpropagation for accommodating much deeper models is another direction we have started working on. We will cite and remark to these aspects in the discussion of a revised version for the manuscript that we are preparing until the end of the rebuttal period.
>
> Meanwhile, what we tried to do, in response to the review request, was to identify and report other in-literature results on the same datasets where comparable sized models (num of parameters), without asynchronous processing were used. The goal being to provide a reference for evaluating how sensible our scores are. These results are summarised in the table below for accuracy.
>
> |                                          | NMNIST | SHD   | DVS gesture |
> |:---:|:---:|:---:|:---:|
> | Bouanane et al. 2023       |  0.976    | 0.772  |                      |
> | Liu et al. 2023                   |               | 0.793  |                      |
> | He et al. 2020                   | 0.983     |            |  0.868           |
> | Ours synchronous            | 0.949     | 0.783  |  0.739           |
> | Ours asynchronous (MS) | 0.963     | 0.816  |   0.856          |
>
> We note that while the depth of the models represented may differ, the number of parameters (model capacity) is comparable. Overall, we think that our reported results are comparable and within the range of what is reported in the literature, for the choice of models and baselines we used to demonstrate the effects of asynchronous processing (we opted always for model than 2 hidden layer deep models, since asynchrony typically unfolds after 2 hidden layers of depth).
>
> Regarding spike sparsity Bouanane et al 2023, and Liu et al 2023 report sparsity metric but it is not clear how they have normalized them (per timestep, per inference, or per the number of data points in the testset). So a tabular comparison is not meaningful/possible but the papers give an indicative impression we hope for comparison across the datasets.
>
> M. S. Bouanane, D. Cherifi, E. Chicca, L. Khacef. Impact of spiking neurons leakages and network recurrences on event-based spatio-temporal pattern recognition. https://doi.org/10.3389/fnins.2023.1244675
>
> S. Liu, V. C. H. Leung, P. L. Dragotti. First-spike coding promotes accurate and efficient spiking neural networks for discrete events with rich temporal structures. https://doi.org/10.3389/fnins.2023.1266003
>
> W. He, Y.J. Wu, L. Deng, G. Li, H. Wang, Y. Tian, W. Ding, W. Wang, Y. Xie. Comparing SNNs and RNNs on Neuromorphic Vision Datasets: Similarities and Differences. https://doi.org/10.1016/j.neunet.2020.08.001

---

> > ### Author Response · Authors · 2024-11-21
> > **part 2 of the answer**
> >
> > Here is also some overview of the papers and the numbers we extracted from them:
> >
> > Bouanane et al. 2023 evaluates the performance of various parameterizations in SNNs on the N-MNIST and SHD datasets. Their models are wider but shallower than ours, resulting in approximately three times the number of parameters for N-MNIST and a comparable number of parameters for SHD.
> >
> > Liu et al. 2023 evaluates the effectiveness of a temporal error versus rate-based error for SHD and DVS gesture. They use more complex neuron models than us (CUBA-LIF and AdLIF). We only consider the results on rate-based error encoding. For SHD, they use a similarly sized model to ours. For DVS gesture, they use a much more sophisticated CNN. Only the SHD results are included in our table.
> >
> > He et al. 2020 attempts an exhaustive comparison between SNNs and recurrent ANNs. On N-MNIST and DVS gesture they trained a shallower but much wider model with more trainable parameters than ours (7.6x for N-MNIST, 3.5x for DVS gesture). They also cover an even larger and more sophisticated CNN for DVS gesture, which has not been included in the table.

---

> ### Comment · Reviewer_z8jo · 2024-11-28
> **Thanks for your response**
>
> I want to thank the authors for their response, I will keep the score.

---

> > ### Author Response · Authors · 2024-11-30
> >
> > Thanks you. This is greatly appreciated.

---

### Official Review · Reviewer_FREL · 2024-11-03

**Soundness:** 4
**Presentation:** 2
**Contribution:** 3
**Rating:** 6
**Confidence:** 3

**Summary:**

Current spike neural networks (SNNs) must compute and integrate all presynaptic currents from the previous layer before performing calculations for the neurons in the next layer. This dependence on layer synchronization deviates significantly from the original intention of asynchronous SNN design. Ideally, neurons should be able to emit and receive spike currents at any time and at any location within the network. In this paper, the authors address this issue and explore potential solutions. They propose a generalized approach for gradient training that allows for scheduling strategies using asynchronously processing neurons. Experiments demonstrate that this method can save energy and improve latency under asynchronous processing.

The authors highlight a key issue within the SNN research community: most SNNs are layer-synchronized, i.e., time-driven, rather than achieving the original goal of implementing asynchronous event-driven mechanisms. The authors discuss and conduct some experiments on asynchronous networks, and the results are intriguing.

At this stage, my rating is "6: Marginally above acceptance threshold." I would be very willing to raise my score if the authors could address and clarify the following issues.

**Strengths:**

1. **Importance of the Research Content**: Although the original intention of SNN research was for asynchronous computation, most current SNN work is time-driven (i.e., layer-synchronized as mentioned in the paper). However, the asynchronous design of SNNs is crucial for applications in event-driven neuromorphic processors. This paper could contribute significantly to the SNN research community.

2. **Novelty of the Results**: The experiments presented in the paper showcase many intriguing advantages of the asynchronous design, such as enhanced neuronal activity while the overall network remains sparser (Figure 2); key information flowing freely during asynchronous inference (Figure 3); and reduced inference latency under asynchronous network processing (Figure 4). These results provide strong support for the future development of asynchronous computation.

**Weaknesses:**

1. **Accuracy of Unlayered Method**: I reviewed the accuracy of the Unlayered method (Table 2), and it generally falls below that of the traditional Layered method. What is the network architecture of the Layered methods compared in Table 2? Can the performance of the Unlayered method be demonstrated using the network architecture from this work [1]? Because these networks are more used by SNN researchers, it would be more convincing to compare the methods in the same network structure.
2. **Network Sparsity and Energy Consumption**: The paper presents many instances of network sparsity; however, the neuronal activity has also increased, and these two factors have opposing effects on energy consumption. Which of these factors is dominant?   Could experiments be conducted to analyze the energy trade-off between the increased neuronal activity and network sparsity, further investigating the impact of each factor on energy consumption within the asynchronous framework? Table 3 shows the energy efficiency of asynchronous computation, but there is no significant reduction, and it does not even decrease by an order of magnitude. I find this outcome somewhat unsatisfactory. Could the authors provide some clarification?

[1]: Stsc-snn: Spatio-temporal synaptic connection with temporal convolution and attention for spiking neural networks, Frontiers in Neuroscience, 2022

**Questions:**

1. **Input Encoding in Asynchronous Processing Framework**: How is event data encoded as input into the network within the asynchronous processing framework? Although lines 210-215 provide an explanation, I am still unclear about the form of the input data. Could the authors provide some equations for clarification? Perhaps an example with a single input event could be illustrated step-by-step, or a small code snippet could be submitted to demonstrate how a single input event is encoded and processed within the asynchronous framework.

2. **UNLAYERED BACKPROPAGATION**: In UNLAYERED BACKPROPAGATION, due to the dynamic nature of asynchronous processing, the backpropagation may occur across layers rather than layer-by-layer in a chained manner. Does this training approach lead to faster convergence during network training?

---

> ### Author Response · Authors · 2024-11-21
> **Answer to first comment**
>
> NB. references to sections and line numbers cited in this and the following answers refer to the original version of the manuscript.
>
> Would like to thank the reviewer for the remarks and the opportunity to defend our work, in order to make it better understandable.
>
> > the accuracy of the Unlayered method (Table 2), and it generally falls below that of the traditional Layered method. What is the network architecture of the Layered methods compared in Table 2?
>
> The models in these comparisons have exactly the same topology/structure and capacity (number or parameters) as the baseline, and were trained with the same loss (cross-entropy). This makes the comparison in our eyes as fair as it can be, and devoid of other factors influencing the results (e.g. different topology, different depth, more complex neuron model, different network block-structure) apart from the asynchronous training. Our goal is to isolate, and amass information about the phenomenon and effects of asynchronous processing, and how to accommodate it in training.
>
> The details of the topology and training configuration per each dataset is reported in A.6.1 (for Table 2) and A.10 for the deeper VGG in section 4.6. The choice of the topology, as we remark in section 4.1, is such that each model has more than 2 hidden layers, because with up to 2 hidden layers, asynchrony barely unfolds even without any layer synchronization. The adverse effects (of synchronous training with asynchronous inference) become more and more pronounced as the depth increases.
>
> If one compares the reported accuracy to the SoA accuracy for these datasets, we acknowledge that higher scores have been reported in the literature. For example for SHD, higher accuracies have been reported in literature by using models with complex multi-compartment neurons, synaptic delays, batch-norm, and other structures (ultra-high parameter capacity models), but we do not see what purpose it would serve to compare against such models. Apart from being complex to implement in our simulation environment, it would necessitate a thorough ablation study afterward to isolate the part of the accuracy that is merited to “each trick” versus the merits of asynchronous execution.
>
> It is important to emphasise that the primary message we aim to convey is not solely about achieving high accuracy, but rather demonstrating competitive or recovered accuracy alongside significant improvements in energy and latency efficiency. This aligns with the premises of neuroscience in spiking neural networks (SNNs), where asynchronous processing offers unique advantages in these aspects.

---

> > ### Author Response · Authors · 2024-11-21
> > **Answer to second comment**
> >
> > > Can the performance of the Unlayered method be demonstrated using the network architecture from this work [1]? Because these networks are more used by SNN researchers, it would be more convincing to compare the methods in the same network structure.
> >
> > Our motivation for this study was to work on type with models that are (or can be) readily deployed on current neuromorphic accelerators, so that in the future we can also perform in-vivo on-hardware measurements (particularly for energy), and so experimentation was scoped with hardware deployability of models in-mind (on platforms such as (Imec) Seneca and uBrain, (Intel) Loihi 1/2, (SynSense) Speck, SpiNNaker, and other similar). Models with non-common structures such as [1], typically lack support on these platforms.
> >
> > The architectural approach of the model in [1] follows the philosophy of the attentional mechanism of transformers, to rid of recurrencies and sequential processing. The caveat of this architecture is that it by-design requires synchronisation points inside the network for the element-wise gating multiplications, which as is (i.e. without architectural ramifications) is not compatible with, and not expected to benefit much from, end-to-end asynchronous processing. Also, like transformers, computationally (energy-wise) it must be more expensive than “normal” RNNs/SNNs.
> >
> > However, we do agree that extending our work to more complex model structures (and quantifying the benefits and tradeoffs), especially the sort of represented by [1] or more suitably the RWKVs, is important, and is part of our agenda for follow-up research (we will add a relevant remark also in the revised version of the manuscript that we are preparing).
> >
> > Additionally, we started an attempt to implement [1] in our simulation environment and if attainable to obtain preliminary/indicative results (in the limited time of the rebuttal), we will include them here and in the supplementary material or the appendix of the paper. We prefer however to keep a clear focus and crisp message in the main text on what asynchrony buys us, and why we should not neglect it as a design tenet of SNNs that are deployable in neuromorphic chips. Our emphasis, not being on accuracy in isolation, but in combination with efficiency (energy/latency).

---

> > > ### Author Response · Authors · 2024-11-21
> > > **Answer to third comment**
> > >
> > > > The paper presents many instances of network sparsity; however, the neuronal activity has also increased, and these two factors have opposing effects on energy consumption. Which of these factors is dominant? Could experiments be conducted to analyze the energy trade-off between the increased neuronal activity and network sparsity, further investigating the impact of each factor on energy consumption within the asynchronous framework?
> > >
> > > We believe that the requested result/experiment is captured at the bottom part of Figure 2.
> > >
> > > To explain/clarify maybe a bit more. The overall neuronal activity has increased by virtue of  evaluating the membrane threshold upon arrival of every current independently (fidel to biology). **This creates the temporal dynamics for asynchronous processing inside the network, whereas most work currently on SNNs only consider the temporal dynamics in the external stimulus**.
> > >
> > > Moreover, in the absence of layer synchonization barriers, some of this spike activity propagates fast to the output, leading to inference decision before other neurons generate new spikes and before the entire already generated spike activity in the network is consumed (processed) in all layers. Because **energy consumption is due to only the spike activity processed** (transactions with the memory) and not the total activity generated (or would be generated), if processing terminates as soon as a decision is made at the output, there is energy (and latency) saving.
> > >
> > > This phenomenon which is empirically observed in this work is in agreement with the theories of time-to-first-spike, rank-order, and N-of-M encodings, but we are able to produce it with rate codes too. The quantification of this result is in essence at the bottom part of Figure 2 (comparing the on-output and on-spiking-done conditions, meaning termination of inference as soon as output neurons get excited versus waiting for all activity to be propagated to the output), which then lead to the quantifications in Table 3 (normalised per inference).

---

> > > > ### Author Response · Authors · 2024-11-21
> > > > **Answer to fourth comment**
> > > >
> > > > > Table 3 shows the energy efficiency of asynchronous computation, but there is no significant reduction, and it does not even decrease by an order of magnitude. I find this outcome somewhat unsatisfactory. Could the authors provide some clarification?
> > > >
> > > > Table 3 reports indicative energy consumption numbers for one neuromorphic accelerator resulting from asynchronous processing of the portion of events that lead up to prediction. This reduction is by a factor of 0.5 for topologies of depth 3 (topologies in table 6 in supplementary material), and it is a function of the depth of the model. As model depth increases the reduction in energy becomes bigger because it depends primarily on the spikes processed, not generated. The deeper and wider the model the more spikes are generated in total (all other aspects being equal), but only a small percentage of them are processed until prediction is ready under asynchronous processing (and the right scheduling policy plays a role here of course). So in other words irrespective of total activation density (or sparsity), the bulk of the energy reduction results from the percentage of them that gets processed.
> > > >
> > > > But there is more to energy reduction, that what we report in the paper. In analog neuromorphic accelerators the number of spikes processed is a fidel measure of the overall energy consumption. However, in a digital accelerator the energy cost is not only due to the spikes processed (memory I/O for synaptic operations) but also due to the leakage of the memory, which in turn is a function of the time the circuit is on (latency of inference). In this case reducing the latency of inference (section 4.5) brings another significant amount of energy saving, which however is difficult to quantify at the algorithm level because it depends on several hardware factors. Among others the (CMOS) technology node used and the clocking frequency of the accelerator. And on top of that, our latency results are worst-case calculations based on sequential processing of the spikes. This overall makes this unaccounted (in the paper) component of energy reduction hard to assess without actual on-hardware measurements, but it is fair to expect that the combination of the two energy components can easily reach an order of magnitude or more. (This is also a planned next step).
> > > >
> > > > If you think this hardware-related discussion is useful for appreciating the value of the work, we can include it in a dedicated section in the appendix of the paper.

---

> > > > > ### Author Response · Authors · 2024-11-21
> > > > > **Answer to fifth comment**
> > > > >
> > > > > > Input Encoding in Asynchronous Processing Framework: How is event data encoded as input into the network within the asynchronous processing framework?
> > > > >
> > > > > We have included a figure to the supplementary matterial (figure 1), that originates from Hagenaars et al. 2021 (ref below). The temporal stream of input events is discretized in time-bins. The time-bins can be of fixed time length or fixed total number of events in them. Each time-bin constructs a time-frame, where pixel position (input feature) counts accumulated events in the respective time-interval (one can also think of them as IF neurons that have integrated in their membrane a time-bin’s worth of events). The time-frames are fed to the network subsequently in discrete timesteps. This framing approach is one of the commonplace input encodings in the literature.
> > > > >
> > > > > Hagenaars, J., Paredes-Vallés, F., & De Croon, G. (2021). Self-supervised learning of event-based optical flow with spiking neural networks. Advances in Neural Information Processing Systems, 34, 7167-7179. (https://arxiv.org/abs/2106.01862)

---

> > > > > > ### Author Response · Authors · 2024-11-21
> > > > > > **Answer to sixth comment**
> > > > > >
> > > > > > > the backpropagation may occur across layers rather than layer-by-layer in a chained manner. Does this training approach lead to faster convergence during network training?
> > > > > >
> > > > > > We are including a new section in the appendix of the paper (revised version that we are preparing) for illustrating the training error curves of our main experiments, and then also for the CIFAR-10 dataset experiment we will add an additional curve to the already existing figure that shows the convergence of the layered model.
> > > > > >
> > > > > > Overall the general observation is not conclusive on whether asynchronous training converges faster or earlier or to smaller error, but rather depends on several factors such as the F-group size, the scheduling policy, the number of timesteps that the input is divided across, and very likely the difficulty of the task.

---

### Author Response · Authors · 2024-11-22
**revised version of the paper (and supplementary materials)**

We have uploaded a revised version of the manuscript with the changes/additions that we discussed in the responses to the reviewers so far (and the new supplementary materials).

---

### Meta-Review · Area_Chair_Ym2X · 2024-12-17

**Metareview:**

This paper examines the possibility of using asynchronous processing for spiking neural networks (SNNs). In brief, the authors are interested in eliminating the constraint that is often imposed in multi-layer SNNs of having every cell on a layer accumulate its inputs and spike (or not) at some clocked time. The authors are interested in relaxing this constraint as it could potentially reduce the latency for inference and increase the energy efficiency. There are several challenges with asynchronous processing, though, one of which is that standard backprop is not well suited to training in the asynchronous regime. The authors propose a solution for this (a form of backrop adapted for asynchronous events) and they explore various scheduling strategies for asynchronous SNNs. They claim that these strategies improve the accuracy of asynchronous networks, putting them on par or better than synchronous networks, and that they reduce the energy footprint of inference.

The strengths of this paper are that it is exploring an important, but under-examined issue in SNNs, and provides some novel solutions. The weaknesses are that the clarity of the paper could be much improved, and the actual improvements provided by the solutions on offer are fairly limited - the authors find that their asynchronous approach scales poorly in terms of both accuracy and efficiency (e.g. it does not work on VGG models). As such, the claims on performance and efficiency improvements only apply in limited, relatively toy models, and do not appear to be relevant to the ML community more broadly. This paper is really best framed as an initial exploration of the limitations of asynchronous processing. In fairness, the authors try to be clear about this with their choice of title. However, arguably, the high bar for acceptance at ICLR requires more than a well-executed study of limitations in a class of model.

Given these considerations, a decision of reject was reached.

**Additional Comments On Reviewer Discussion:**

The discussion was overall fine, though the most critical reviewer did not respond to the authors rebuttal. They did say in discussion that they were not satisfied with the authors' responses, though.

Importantly, however, the AC did not take this reviewer's comments as the major consideration when rendering their decision. The decision was reached based on a mixture of the reviews and the AC's assessment of the concerns raised and whether they were truly attended to in the rebuttals.

---

### Decision · Program_Chairs · 2025-01-22

Reject